# Open Issues for Protein Function Assignment in *Haloferax volcanii* and Other Halophilic Archaea

**DOI:** 10.3390/genes12070963

**Published:** 2021-06-24

**Authors:** Friedhelm Pfeiffer, Mike Dyall-Smith

**Affiliations:** 1Computational Biology Group, Max-Planck-Institute of Biochemistry, 82152 Martinsried, Germany; mike.dyallsmith@gmail.com; 2Veterinary Biosciences, Faculty of Veterinary and Agricultural Sciences, University of Melbourne, Parkville 3010, Australia

**Keywords:** haloarchaea, genome annotation, Gold Standard Protein, *Haloferax volcanii*, annotation error

## Abstract

Background: Annotation ambiguities and annotation errors are a general challenge in genomics. While a reliable protein function assignment can be obtained by experimental characterization, this is expensive and time-consuming, and the number of such Gold Standard Proteins (GSP) with experimental support remains very low compared to proteins annotated by sequence homology, usually through automated pipelines. Even a GSP may give a misleading assignment when used as a reference: the homolog may be close enough to support isofunctionality, but the substrate of the GSP is absent from the species being annotated. In such cases, the enzymes cannot be isofunctional. Here, we examined a variety of such issues in halophilic archaea (class Halobacteria), with a strong focus on the model haloarchaeon *Haloferax volcanii*. Results: Annotated proteins of *Hfx. volcanii* were identified for which public databases tend to assign a function that is probably incorrect. In some cases, an alternative, probably correct, function can be predicted or inferred from the available evidence, but this has not been adopted by public databases because experimental validation is lacking. In other cases, a probably invalid specific function is predicted by homology, and while there is evidence that this assigned function is unlikely, the true function remains elusive. We listed 50 of those cases, each with detailed background information, so that a conclusion about the most likely biological function can be drawn. For reasons of brevity and comprehension, only the key aspects are listed in the main text, with detailed information being provided in a corresponding section of the Supplementary Materials. Conclusions: Compiling, describing and summarizing these open annotation issues and functional predictions will benefit the scientific community in the general effort to improve the evaluation of protein function assignments and more thoroughly detail them. By highlighting the gaps and likely annotation errors currently in the databases, we hope this study will provide a framework for experimentalists to systematically confirm (or disprove) our function predictions or to uncover yet more unexpected functions.

## 1. Introduction

*Haloferax volcanii* is a model organism for halophilic archaea [1,2,3,4,5,6], for which an elaborate set of genetic tools has been developed [7,8,9]. Its genome has been sequenced and carefully annotated [1,10,11]. A plethora of biological aspects have been successfully tackled in this species, with examples including DNA replication [4]; cell division and cell shape [12,13,14,15,16]; metabolism [17,18,19,20,21,22,23,24,25]; protein secretion [26,27,28,29]; motility and biofilms [30,31,32,33,34,35]; mating [36]; signaling [37]; virus defense [38]; proteolysis [39,40,41,42,43,44]; posttranslational modification (N-glycosylation; SAMPylation) [45,46,47,48,49,50]; gene regulation [21,25,51,52,53,54,55]; microproteins [56,57,58] and small noncoding RNAs (sRNAs) [59,60,61,62].

Genome annotations are frequently compromised by annotation errors [11,63,64,65]. Many of these errors are caused by an invalid annotation transfer between presumed homologs, which, once introduced, are further spread by annotation robots. This problem can be partially overcome by using a Gold Standard Protein (GSP)-based annotation strategy [11]. Since the GSP has itself been subjected to an experimental analysis, its annotation cannot be caused by an invalid annotation transfer process. The GSP strategy was already applied to a detailed analysis of the metabolism of halophilic archaea [66]. However, with a decreasing level of sequence identity, the assumption of isofunctionality becomes increasingly uncertain. Although this may be counterbalanced by additional evidence, e.g., gene clustering, experimental confirmation would be the best option for validation of the annotation.

There are additional and much more subtle genome annotation problems. In some cases, GSPs are true homologs, and the annotated function in the database is correct. Nevertheless, the biological context in the query organism makes it unlikely that the homologs are isofunctional, e.g., when the substrate of the GSP is lacking in the query organism. Additionally, paralogs may have distinct but related functions that cannot be assigned by a sequence analysis but may be assigned based on phylogenetic considerations. Here, again, experimental confirmation is the preferred option for validation of the annotation. A lack of experimental confirmation may keep high-level databases like KEGG or the SwissProt section of UniProt from adopting assignments based on well-supported bioinformatic analyses, so that the database entries continue to provide information that is probably incorrect. We refer to annotation problems in these databases solely to underscore that the biological issues raised by us are far from trivial. There is no intention to question the exceedingly high quality of the SwissProt and KEGG databases [67,68] and their tremendous value for the scientific community. We have actively supported them by providing feedback and encourage others to do the same, e.g., with the recently implemented “Add a publication” functionality in the UniProt entries that allows users to connect a protein to a publication that describes its experimental characterization (https://community.uniprot.org/bbsub/bbsubinfo.html).

In this study, we describe a number of annotation issues for haloarchaea, with a strong emphasis on *Hfx. volcanii*. We denote such cases as “open annotation issues” with the hope of attracting members of the *Haloferax* community and other groups working with halophilic archaea to apply experimental analyses to elucidate the true function(s) of these proteins. This will increase the number of Gold Standard Proteins that originate from *Hfx. volcanii* or other haloarchaea, reduce genome annotation ambiguities and perhaps uncover novel metabolic processes.

## 2. Materials and Methods

### 2.1. Curation of Genome Annotation and Gold Standard Protein Identification

The Gold Standard Protein-based curation of haloarchaeal genomes has been described previously [11] (see, also, next paragraph). Since then, a systematic comparison to the KEGG data was performed for a subset of the curated genomes [69]. The *Hfx. volcanii* genome annotation is continuously scrutinized, especially when a closely related genome is annotated [70]. 

In brief, the core rule of Gold Standard Protein-based genome annotation is to assign a specific function only when a homologous protein has been confirmed experimentally to have this function. Two types of data must be available for that homolog: (a) a reference describing the experimental characterization and (b) an entry in a sequence database, so that the level of sequence similarity can be determined. The decision on whether isofunctionality can be assumed at this level of sequence similarity and, thus, if the annotation can be transferred represents an informed prediction by the annotator based on the available evidence. This decision may be taken only once for a set of closely related orthologs, such as those from halophilic archaea.

### 2.2. Additional Bioinformatics Tools

The key databases were UniProtKB/SwissProt [68], InterPro [71], KEGG [67] and OrthoDB [72]. The SyntTax server was used for inspecting the conservation of the gene neighborhood [73]. As general tools, the BLAST suite of programs [74,75] was used for sequence comparisons. 

## 3. Results

The open issues are organized below under Section 3.1, the respiratory chain and oxidative decarboxylation; Section 3.2, amino acid metabolism; Section 3.3, heme and cobalamin biosynthesis; Section 3.4, coenzyme F420; Section 3.5, tetrahydrofolate as opposed to methanopterin; Section 3.6, NAD and riboflavin; Section 3.7, lipid metabolism; Section 3.8, genetic information processing and Section 3.9, stand-alone (miscellaneous) cases. We collected this set of open annotation issues during our continuous efforts to keep the *Hfx. volcanii* genome up-to-date since its initial publication in 2010 [1]. Not covered in this study are enigmatic reactions and pathways (e.g., archaeal signal peptidase II or the haloarchaeal O-glycosylation pathway) for which no support from experimentally characterized homologs (GSP proteins) is available.

### 3.1. The Respiratory Chain and Oxidative Decarboxylation

In the respiratory chain, the coenzymes that were reduced during catabolism (e.g., glycolysis) are reoxidized, with the energy being saved as an ion gradient. The textbook examples of a respiratory chain are the five mitochondrial complexes [76,77]: complex I (NADH dehydrogenase), complex II (succinate dehydrogenase), complex III (cytochrome bc_1_ complex), complex IV (cytochrome-c oxidase as a prototype for a terminal oxidase) and complex V (F-type ATP synthase). In mitochondria, a significant part of the NADH that feeds into the respiratory chain originates from oxidative decarboxylation: the conversion of pyruvate to acetyl-CoA by the pyruvate dehydrogenase complex and conversion of α-ketoglutarate to succinyl-CoA by the homologous 2-oxoglutarate dehydrogenase complex. While complexes I and II transfer reducing elements to a lipid-embedded two-electron carrier (ubiquinone), the bc_1_ complex transfers the electrons to the one-electron carrier cytochrome-c, a heme (and, thus, iron) protein, which then transfers electrons to the terminal oxidase.

Bacteria like *Escherichia coli* and *Paracoccus denitrificans* have related complexes and enzymes: NADH dehydrogenase (encoded by the *nuo* operon), succinate dehydrogenase (encoded by *sdhABCD*) and the related fumarate reductase (encoded by *frdABCD*) [78], several terminal oxidases (e.g., products of *cyoABCDE* and *cydABC*) and an F-type ATP synthase (encoded by *atp* genes). *E. coli* lacks a bc_1_ complex, which, however, occurs in *Paracoccus denitrificans* [79]. *E. coli* contains the canonical complexes of oxidative decarboxylation (the pyruvate dehydrogenase complex, encoded by *aceEF*+*lpdA*, and the 2-oxoglutarate dehydrogenase complex, encoded by *sucAB*+*lpdA*).

The respiratory chain of *Hfx. volcanii* and other haloarchaea deviates considerably from those of mitochondria and bacteria such as *Paracoccus* and *E. coli* (reviewed by [80]), and a number of questions remain unresolved. We focus on the equivalents of complexes I, III and IV, because these have unresolved issues. We also cover some aspects relevant for the NADH levels (oxidative decarboxylation enzymes and type II NADH dehydrogenase). We do not cover complexes that have already been studied in haloarchaea: complex II (succinate dehydrogenase) [81,82,83] and complex V (ATP synthase) [84,85]. 

(a) In haloarchaea, oxidative decarboxylation is not linked to the reduction of NAD to NADH but to the reduction of a ferredoxin (encoded by *fdx*, e.g., OE_4217R, HVO_2995), which has a redox potential similar to that of the NAD/NADH pair [86]. The enzymes for oxidative decarboxylation are pyruvate–ferredoxin oxidoreductase (*porAB*, e.g., OE_2623R/2622R and HVO_1305/1304) and 2-oxoglutarate–ferredoxin oxidoreductase (*korAB*, e.g., OE_1711R/1710R and HVO_0888/0887), and these have been characterized from *Halobacterium salinarum* [87,88,89].

(b) It is yet unresolved how ferredoxin Fdx is reoxidized, but this might be achieved by the Nuo complex. This ferredoxin may well be involved in additional metabolic processes. In *Hfx. volcanii*, ferredoxin Fdx (HVO_2995) plays an essential role in nitrate assimilation [90]. However, in *Hbt. salinarum*, this metabolic process for Fdx reoxidation does not exist. 

(c) The *nuo* cluster of haloarchaea resembles that of *E. coli*, a type I NADH dehydrogenase, with the genes and gene order highly conserved and just a few domain fissions and fusions. However, haloarchaea lack NuoEFG [91], which is a subcomplex that mediates interaction with NADH [92,93]. Thus, the haloarchaeal *nuo* complex is unlikely to function as NADH dehydrogenase, despite its annotation as such in KEGG (as of April 2021).

(d) Other catabolic enzymes generate NADH, which must also be reoxidized. Based on inhibitor studies, NADH is not reoxidized by a type I but, rather, by a type II NADH dehydrogenase in *Hbt. salinarum* [82]. A tentative gene assignment has been made for *Natronomonas pharaonis* [66]. However, for reasons detailed in Appendix A, this assignment is highly questionable, so this issue calls for an experimental analysis.

(e) About one-third of the haloarchaea, especially the *Natrialbales*, do not code for a complex III equivalent (the cytochrome bc_1_ complex encoded by *petABC*), according to OrthoDB analysis. The bc_1_ complex is required to transfer electrons from the lipid-embedded two-electron carrier (menaquinone in haloarchaea) to the one-electron carrier associated with terminal oxidases (probably halocyanin). How electrons flow in the absence of a complex III equivalent is currently unresolved.

The haloarchaeal *petABC* genes resemble those of the chloroplast b6-f complex rather than those of the mitochondrial bc_1_ complex (see Appendix A for more details).

(f) A bc cytochrome was purified from *Nmn. pharaonis*, but with an atypical 1:1 ratio between the b-type and c-type hemes [81]. The complex is heterodimeric, with subunits of 18 kDa and 14 kDa. The 18-kDa subunit carries the covalently attached heme group [81]. An attempt was made to identify the genes coding for these subunits [94] (for details, see Appendix A). Two approaches were used to obtain protein sequence data, one being the N-terminal protein sequencing of the two subunits extracted from a SDS-polyacrylamide gel. In the other attempt, peptides from the purified complex were separated by HPLC, and a peptide which absorbed at 280 nm (protein), as well as 400 nm (heme), was isolated. Absorption at 400 nm clearly indicates that the isolated peptide contains a covalently attached heme group. The sequences from the two approaches overlapped and resulted in a contiguous sequence of 41 aa, with only the penultimate position remaining undefined [94]. Based on this information, a PCR probe was generated (designated “cyt-C Sonde”) that allowed the gene to be identified and sequenced, including its genomic neighborhood. It turned out that the genes coding for the four subunits of succinate dehydrogenase (*sdhCDBA*) were isolated. The obtained protein sequence corresponds to the N-terminal region of *sdhD* (with the initiator methionine cleaved off) and only two sequence discrepancies, in addition to the unresolved penultimate residue.

In the PhD thesis [94], this unambiguous result was rated to be a failure (and the data were never formally published). The reason is that SdhD is free of cysteine residues, while standard textbooks state that a pair of cysteines is required for covalent heme attachment [95]. The lack of the required cysteine pair was taken to indicate that the results were incorrect and that the identified genes did not encode the cytochrome bc that the study was seeking [94]. In contrast, we speculate that the results were completely correct, despite being in conflict with the cysteine pair paradigm. In our opinion, a paradigm shift is required. The obtained results call for a yet-unanticipated novel mode of covalent heme attachment, exemplified by the 18-kDa subunit of *Natronomonas* succinate dehydrogenase subunit SdhD. It should be noted that the 41-aa protein sequence, which was obtained, turned out to contain three histidine residues upon translation of the gene, but none of these were detected upon Edman degradation.

In *Halobacterium*, a small c-type cytochrome was purified (cytochrome c_552_, 14.1 kDa) [96]. Heme staining after SDS-PAGE indicated a covalent heme attachment, but no sequence or composition data were reported, so that it was not possible to identify the protein based on the available information. We speculate that the *Halobacterium* cytochrome c_552_ also represents SdhD (as detailed in Appendix A). In that case, the proposed novel type of covalent heme attachment would not be restricted to *Nmn. pharaonis* but might be a general property of haloarchaea. This would also solve the “*Halobacterium* paradox” [95].

(g) The haloarchaeal one-electron carrier is the copper protein halocyanin rather than the iron-containing heme protein cytochrome-c. A halocyanin from *Nmn. pharaonis* (NP_3954A) was characterized, including its redox potential [97,98,99]. A gene fusion supports the close connection of a halocyanin with a subunit of a terminal oxidase. For further details, see Appendix A.

(h) Terminal oxidases are highly diverse in haloarchaea, and we restricted our analysis to three species (*Nmn. pharaonis*, *Hfx. volcanii* and *Hbt. salinarum*), because in each of these, at least one terminal oxidase has been experimentally studied (Table 1). The details are described in Appendix A with subunits of all analyzed terminal oxidases listed in Appendix A.

(i) NAD-dependent oxidative decarboxylation is a canonical reaction to convert pyruvate into acetyl-CoA and α-ketoglutarate into succinyl-CoA. In haloarchaea, the conversion of pyruvate to acetyl-CoA and α-ketoglutarate to succinyl-CoA is dependent on ferredoxin, not on NAD (see above). Nevertheless, most haloarchaeal genomes also code for homologs of enzymes catalyzing NAD-dependent oxidative decarboxylation, such as the *E. coli* pyruvate dehydrogenase complex. In most cases, the substrates could not be identified, an exception being a paralog involved in isoleucine catabolism [116]. In several cases, the enzymes were found not to show catalytic activity with pyruvate or α-ketoglutarate (see Appendix A for details). Additionally, a conditional lethal *porAB* mutant was unable to grow on glucose or pyruvate, thus excluding that alternative enzymes for the conversion of pyruvate to acetyl-CoA exist in *Hfx. volcanii* [22]. Nonetheless, despite experimental results to the contrary, pyruvate has been assigned as a substrate for some of the homologs of the pyruvate dehydrogenase complex in KEGG (as of April 2021). 

### 3.2. Amino Acid Metabolism

While most amino acid biosynthesis and degradation pathways can be reliably reconstructed, a few open issues remain, which are discussed below.

(a) The first and last steps of arginine biosynthesis deal with blocking and unblocking of the α-amino group of the substrate (glutamate) and a product intermediate (ornithine). As detailed in Appendix A, it is highly likely that glutamate is attached to the γ-carboxyl group of a carrier protein, and ornithine is released from that carrier protein. This is based on characterized proteins from *Thermus thermophilus* [124], *Thermococcus kodakarensis* [125] and *Sulfolobus acidocaldarius* [126]. The assignment is strongly supported by clustering of the arginine biosynthesis genes. Some of the homologs are bifunctional, being involved in arginine biosynthesis but, also, in lysine biosynthesis via the prokaryotic variant of the α-aminoadipate pathway. This ambiguity is not assumed to occur in haloarchaea, which use the diaminopimelate pathway for lysine biosynthesis [127] (see Appendix A for further discussion of this issue).

Expanding the above, we provided full details underlying our reconstruction of arginine and lysine biosynthesis in *Hfx. volcanii* in Table 2.

**Table 2 genes-12-00963-t002:** Proteins with open annotation issues and their Gold Standard Protein homologs (Section 3.2). For a description of this table, see the legend to Table 1.

			Gold Standard Protein						
Section	Code	Gene	isofunc	%seq_id	Locus tag	UniProt	Reference	PMID	Comment
2a	HVO_0047	*argW*	no	54%	TT_C1544	Q72HE5	[128]	25392000	for Arg, not for Lys biosynthesis
2a	HVO_0047 (cont.)		yes/no	39%	Saci_0753	Q4JAQ0			only for Arg, not for Lys biosynthesis
2a	HVO_0047 (cont.)		yes/no	61%	TK0279	Q5JFV9	[125]	27566549	only for Arg, not for Lys biosynthesis
2a	HVO_0046	*argX*	no	44%	TT_C1543	Q72HE6	[124]	19620981	for Arg, not for Lys biosynthesis
2a	HVO_0046 (cont.)		yes	30%	Saci_1621	Q4J8E7			only for Arg, not for Lys biosynthesis
2a	HVO_0046 (cont.)		yes/no	37%	TK0278	Q5JFW0	[125]	27566549	only for Arg, not for Lys biosynthesis
2a	HVO_0044	*argB*	no	41%	TT_C1541	O50147	[124] [128]	19620981 25392000	for Arg, not for Lys biosynthesis
2a	HVO_0044 (cont.)		yes/no	33%	Saci_0751	Q4JAQ2	[126]	23434852	only for Arg, not for Lys biosynthesis
2a	HVO_0044 (cont.)		yes/no	32%	TK0276	Q5JFW2	[125]	27566549	only for Arg, not for Lys biosynthesis
2a	HVO_0045	*argC*	no	48%	TT_C1542	O50146	[124] [129]	19620981 26966182	for Arg, not for Lys biosynthesis
2a	HVO_0045 (cont.)		yes/no	42%	Saci_0750	Q4JAQ3	[126]	23434852	only for Arg, not for Lys biosynthesis
2a	HVO_0045 (cont.)		yes/no	46%	TK0277	Q5JFW1	[125]	27566549	only for Arg, not for Lys biosynthesis
2a	HVO_0043	*argD*	no	45%	TT_C1393	Q93R93	[130]	11489859	for Arg, not for Lys biosynthesis
2a	HVO_0043 (cont.)		yes/no	40%	Saci_0755	Q4JAP8	[126]	23434852	only for Arg, not for Lys biosynthesis
2a	HVO_0043 (cont.)		yes/no	42%	TK0275	Q5JFW3	[125]	27566549	only for Arg, not for Lys biosynthesis
2a	HVO_0042	*argE*	no	36%	TT_C1396	Q8VUS5	[124] [131]	19620981 28720495	for Arg, not for Lys biosynthesis
2a	HVO_0042 (cont.)		yes/no	29%	Saci_0756	Q4JAP7	[126]	23434852	only for Arg, not for Lys biosynthesis
2a	HVO_0042 (cont.)		yes/no	37%	TK0274	Q5JFW4	[125]	27566549	only for Arg, not for Lys biosynthesis
2a	HVO_0041	*argF*	yes	50%	P18186	BSU11250	[132]	4216455	
2a	HVO_0041 (cont.)		yes	47%	OE_5205R	B0R9X3	[133]	7868583	
2a	HVO_0049	*argG*	yes	35%	-	P00966	[134]	8792870	human
2a	HVO_0049 (cont.)		yes	23%	b3172	P0A6E4	[135]	10666579	*E. coli*
2a	HVO_0048	*argH*	yes	38%	MMP0013	O74026	[136]	10220900	
2a	HVO_0008	*lysC*	yes	32%	BSU28470	P08495	[137]	15033471	
2a	HVO_2487	*asd*	yes	51%	MJ0205	Q57658	[138]	16225889	
2a/9e	HVO_1101	*dapA*	yes	45%	PA1010	Q9I4W3	[139]	21396954	
2a	HVO_1100	*dapB*	yes	33%	b0031	P04036	[140]	7893644	
2a	HVO_1099	*dapD*	yes	32%	b0166	P0A9D8	[141]	6365916	
2a	HVO_1096	*dapE*	yes	29%	b2472	P0AED7	[142]	3276674	function supported by gene clustering
2a	HVO_1097	*dapF*	yes	35%	b3809	P0A6K1	[143]	6378903	
2a	HVO_1098	*lysA*	yes	38%	b2838	P00861	[144]	14343156	
2a	HVO_A0634	-	unknown	25%	b2472	P0AED7	[142]	3276674	function assigned to HVO_1096 in *dap* cluster
2b	HVO_0790	*fba2*	special	67%	OE_1472F	B0R334	[145]	25216252	EC 2.2.1.10 activity of OE_1472F not yet confirmed in vitro
2b	HVO_0790 (cont.)		special	45%	MJ0400	Q57843	[146]	15182204	substrate uncertain
2b	HVO_0792	*aroB*	yes	69%	OE_1475F	B0R336	[145]	25216252	OE_1475F only partially characterized
2b	HVO_0792 (cont.)		yes	44%	MJ1249	Q58646	[146]	15182204	
2b	HVO_0602	*aroD1*	yes	44%	OE_1477R	B0R338	[145]	25216252	
2b	HVO_0602 (cont.)		yes	31%	MMP1394	Q6LXF7	[147]	15262931	
2c	HVO_0009	*tnaA*	yes	41%	b3708	P0A853	[148] [149]	2659590 14284727	
2d	HVO_A0559	*hutH*	yes	42%	BSU39350	P10944	[150] [151]	2454913 14066617	
2d	HVO_A0562	*hutU*	yes	62%	BSU39360	P25503	[152]	4990470	
2d	HVO_A0560	*hutI*	yes	42%	BSU39370	P42084	[153]	16990261	
2d	HVO_A0561	*hutG*	yes	33%	BSU39380	P42068	[152]	4990470	
2e	HVO_0431	-	-						no GSP available
2e	HVO_0644	*leuA1*	yes/no	47%	MJ1392	Q58787	[154]	9864346	HVO_0644 monofunc (CimA) or bifunc (CimA+LeuA); MJ1392 CimA
2e	HVO_0644 (cont.)		unclear	44%	MJ1195	Q58595	[155]	9665716	HVO_0644 monofunc (CimA) or bifunc (CimA+LeuA); MJ1195 LeuA
2e/2f	HVO_1510	*leuA2*	yes	47%	MJ1195	Q58595	[155]	9665716	HVO_1510 LeuA; MJ1195 LeuA
2e/2f	HVO_1510 (cont.)		no	41%	MJ1392	Q58787	[154]	9864346	HVO_1510 LeuA MJ1392 CimA
2e	HVO_A0489	*-*	no	31%	MJ1392	Q58787	[154]	9864346	HVO_A0489 general function only; MJ1392 CimA
2e	HVO_A0489 (cont.)		no	30%	MJ1195	Q58595	[155]	9665716	HVO_A0489 general function only; MJ1195 LeuA
2e	HVO_1153	-	-						function unassigned; no GSP

(b) Archaea use a different precursor for aromatic amino acid biosynthesis than the classical pathway. This has been resolved for *Methanocaldococcus jannaschii* and for *Methanococcus maripaludis* [146,156]. However, the initial steps may differ from those reported for *Methanocaldococcus* in that fructose 1,6-bisphosphate, rather than 6-deoxy-5-ketofructose, might be a substrate [145]. Up to now, a clean deletion of the corresponding enzymes and confirmation with in vitro assays has not yet been achieved (for details, see Appendix A).

(c) The gene for tryptophanase (*tpa*) is stringently regulated in *Haloferax*, which is the basis for using its promoter in the toolbox for regulated gene expression [157]. The shutdown of this gene avoids tryptophan degradation when supplies are scarce. Tryptophanase cleaves tryptophan into indole, pyruvate and ammonia. The fate of indole is, however, yet unresolved.

(d) A probable histidine utilization cluster exists, based on the characterized homologs from *Bacillus subtilis*, but has not yet been experimentally verified.

(e) Among the 16 auxotrophic mutants observed in a *Hfx. volcanii* transposon insertion library [9], some could grow only in the presence of one (or several) supplied amino acids. In many cases, the affected genes were known to be involved in the corresponding pathway, but the others may lead to novel function assignments. One affected gene resulted in histidine auxotrophy, and the product of this gene (HVO_0431) is an interesting candidate. The InterPro domain assignment (HAD family hydrolase) fits into the only remaining pathway gap in histidine biosynthesis (histidinol-phosphatase). In this context, it should be noted that the enzyme that catalyzes the preceding reaction (encoded by *hisC*) is part of a highly conserved three-gene operon involved in polar lipid biosynthesis (see below). For details, see Appendix A. One affected gene resulted in isoleucine auxotrophy. The product of this gene (HVO_0644) is currently annotated to catalyze two reactions, one being an early step in isoleucine biosynthesis (EC 2.3.1.182) and the other being the first step after leucine biosynthesis branches off from valine biosynthesis (EC 2.3.3.13) (see below, (f)) (for details, see Appendix A).

(f) *Hfx. volcanii* codes for two paralogs with an attributed function as 2-isopropylmalate synthase (EC 2.3.3.13). This is the first reaction specific to leucine biosynthesis when the pathway branches off valine biosynthesis. One paralog, HVO_0644, is annotated as bifunctional, also catalyzing a chemically similar reaction that is an early step in isoleucine biosynthesis (EC 2.3.1.182). When the gene encoding HVO_0644 is disrupted by transposon integration, cells cannot grow in the absence of isoleucine. It is unclear if the protein is really bifunctional and is really involved in leucine biosynthesis, catalyzing the reaction of EC 2.3.3.13. The other paralog, HVO_1510, belongs to an ortholog set with major problems concerning the start codon assignment. The ortholog set from the 16 genomes listed in Appendix A was analyzed. When only canonical start codons are considered (ATG, GTG and TTG), the orthologs from *Haloferax mediterranei*, *Nmn. pharaonis*, *Natronomonas moolapensis* and *Halohasta litchfieldiae* either lack a long highly conserved N-terminal region or they are disrupted (pseudogenes), being devoid of a potential start codon. The gene from *Hfx. volcanii* has a start codon (GTG) that is consistent with that of *Haloferax gibbonsii* strain LR2-5 (but a GTA in *Hfx. gibbonsii* strain ARA6). In this region, the gene from *Hfx. mediterranei* is closely related but has in-frame stop codons. HVO_1510 is considerably longer than the orthologs from *Haloquadratum walsbyi*, *Haloarcula hispanica* and *Natrialba magadii*. The first alternative start codon for HVO_1510 codes for Met-93. This protein was proteomically identified in three ArcPP datasets [2], and peptides upstream of Met-93 were identified. This gene might be translated from an atypical start codon, either an in-frame CTG or an out-of-frame ATG, which would require ribosomal slippage (for details, see Appendix A). It is tempting to speculate that translation occurs only when leucine is not available.

### 3.3. Coenzymes I: Cobalamin and Heme

The classical heme biosynthesis pathway branches off cobalamin biosynthesis at the level of uroporphyrinogen III. A second pathway exists in bacteria (CPD pathway). Haloarchaea use the alternative heme biosynthesis pathway [158], which has an additional common step with cobalamin biosynthesis, the conversion of uroporphyrinogen III to precorrin-2. For heme biosynthesis, precorrin-2 is converted into siroheme. This pathway was reconstructed [159], except for the iron insertion step. For de novo cobalamin biosynthesis, haloarchaea use the cobalt-early pathway. A key reaction in this pathway variant, catalyzed by CbiG, is cobalt-dependent. Thus, cobalt must be inserted early and is present in all intermediates [160]. Several aspects of heme and cobalamin biosynthesis in haloarchaea have yet to be resolved. This is illustrated in Figure 1.

(a) *Hfx. volcanii* contains two annotated *cbiX* genes. For the reasons detailed in Appendix A, we predict that one is a cobaltochelatase, involved in cobalamin biosynthesis, while the other is a ferrochelatase, responsible for the conversion of precorrin-2 to siroheme in the alternative heme biosynthesis pathway. 

(b) De novo cobalamin biosynthesis has been extensively reconstructed upon curation of the genome annotation [11]. All enzymes of the pathway and their associated GSPs are listed in Table 3. Only two pathway gaps remained, and because these are consecutive, it may be possible that the haloarchaeal pathway is noncanonical and proceeds via a novel biosynthetic intermediate. There are only four genes with yet-unassigned functions in the *Hfx. volcanii* cobalamin gene cluster, and their synteny is well-conserved in the majority of haloarchaeal genomes. Thus, these genes are obvious candidates for filling the pathway gaps (for details, see Appendix A). 

(c) The cobalamin biosynthesis and salvage reactions (those beyond ligand cobyrinate a,c diamide) involve “adenosylation of the corrin ring, attachment of the aminopropanol arm, and assembly of the nucleotide loop that bridges the lower ligand dimethylbenzimidazole and the corrin ring” [161]. The enzymes of these branches of cobalamin biosynthesis and their associated GSPs are listed in Table 3. Only two pathway gaps remain open. For one of these, a candidate was proposed upon a detailed bioinformatic analysis [161] (for further details, see Appendix A).

(d) In the cobalt-late (aerobic) pathway variant, the intermediates are cobalt-free, and cobalt is inserted only late in the pathway. Even though haloarchaea do not use the cobalt-late pathway, so that a late cobaltochelatase is not required, they code for a homolog of the large subunit of a characterized heterotrimeric late cobaltochelatase. The adjacent gene is homologous to small subunits of other chelatases. We speculate that this late cobaltochelatase may be involved in cobalamin salvage. The chelatase has a mosaic subunit structure, as also reported previously [161] (see Appendix A for details).

(e) In the alternative heme biosynthesis pathway, siroheme is decarboxylated to 12,18-didecarboxysiroheme, which is attributed to the proteins encoded by *ahbA* and *ahbB*. These are homologous to each other and are organized as two two-domain proteins. It is unclear if AhbA and AhbB function independently or if they form a complex.

(f) Two of the three heme biosynthesis pathways (AHB and CPD) share a common last step (decarboxylation of Fe-coproporphyrin III to protoheme (heme b)). They use, however, distinct types of enzymes (AHB: *ahbD*, EC 1.3.98.6, adenosylmethionine-dependent heme synthase, a radical SAM enzyme; CPD: *chdC*, EC 1.3.98.5, peroxide-dependent heme synthase). Nearly all haloarchaea contain a *chdC* gene, and two-thirds also contain an *ahbD* gene. *Hfx. volcanii* was shown to use AhbD under anaerobic conditions and ChdC under aerobic conditions [162].

**Table 3 genes-12-00963-t003:** Proteins with open annotation issues and their Gold Standard Protein homologs (Section 3.3). For a description of this table, see the legend to Table 1.

			Gold Standard Protein						
Section	Code	Gene	Isofunc	%seq_id	Locus Tag	UniProt	Reference	PMID	Comment
3a	HVO_B0054	*cbiX1*	yes	30%	-	O87690	[163]	12408752	cobaltochelatase
3a	HVO_B0054 (cont.)		yes	27%	MTH_1397	O27448	[164]	12686546	cobaltochelatase
3a	HVO_1128	*cbiX2*	no	29%	AF0721	O29537	[165]	16835730	cobaltochelatase
3a	HVO_1128 (cont.)		no	28%	MTH_1397	O27448	[164]	12686546	cobaltochelatase
3a	HVO_1128 (cont.)		no	29%	AF0721	O29537	[165]	16835730	cobaltochelatase
3a	NP_0734A	*cbiX3*	-						function unassigned; no GSP; distantly related to paralogs
3a	HVO_2312	*sirC*	yes/no	31%	Mbar_A1461	Q46CH4	[166]	21197080	precorrin-2 DH; no analysis for Fe-chelatase
3a	HVO_2312 (cont.)		yes/no	29%	STM3477	P25924	[167] [168]	14595395 32054833	matches to the N-term domain which is bifunctional as precorrin-2 DH and Fe-chelatase
3a	HVO_2312 (cont.)		yes/no	29%	-	P61818	[163] [169]	12408752 18588505	precorrin-2 DH; devoid of Fe-chelatase activity
3b	HVO_B0061	*cbiL*	no	32%	STM2024	Q05593	[170]	1451790	equivalent reaction on cobalt-free substrate
3b	HVO_B0057	*cbiH2*	yes	45%	-	O87689	[160]	23922391	corresponds to N-term of O87689 which has a C-term extension
3b	HVO_B0057 (cont.)		no	40%	STM2027	Q05590	[171] [172]	9331403 16198574	equivalent reaction on cobalt-free substrate
3b	HVO_B0058	*cbiH1*	special	32%	-	O87689	[160]	23922391	corresponds to N-term of O87689 which has a C-term extension; more distant to O87689 than CbiH2
3b	HVO_B0058 (cont.)		no	30%	STM2027	Q05590	[171] [172]	9331403 16198574	equivalent reaction on cobalt-free substrate
3b	HVO_B0060	*cbiF*	no	40%	STM2029	P0A2G9	[170] [173]	1451790 16866557	equivalent reaction on cobalt-free substrate
3b	HVO_B0060 (cont.)		yes	38%	-	O87686	[160]	23922391	
3b	HVO_B0059	*cbiG*	yes	24%	-	O87687	[160]	23922391	
3b	pathway gap								EC 2.1.1.195
3b	pathway gap								EC 1.3.1.106
3b	HVO_B0062	*cbiT*	yes	36%	-	O87694	[160]	23922391	corresponds to the C-term of bifunctional O87694
3b	HVO_B0048	*cbiE*	yes	28%	-	O87694	[160]	23922391	corresponds to the N-term of bifunctional O87694
3b	HVO_B0049	*cbiC*	yes	33%	-	O87692	[160]	23922391	
3b	HVO_A0487	*cbiA*	no	37%	STM2035	P29946	[174]	15311923	equivalent reaction on cobalt-free substrate
3b	HVO_B0052	-	-						function unassigned; no GSP
3b	HVO_B0053	-	-						function unassigned; no GSP
3b	HVO_B0055	-	-						function unassigned; no GSP
3b	HVO_B0056	-	-						function unassigned; no GSP
3c	HVO_A0488	*cobA*	yes	31%	MM_3138	Q8PSE1	[175]	16672609	
3c	HVO_A0488 (cont.)		yes	30%	STM1718	P31570	[176]	12080060	
3c	HVO_2395	*pduO*	yes	37%	-	Q9XDN2	[177]	11160088	PduO and CobA are isofunctional; In Q9XDN2, the PduO domain (N-term) is fused to a DUF336 domain
3c	HVO_A0553	*cbiP*	yes	63%	VNG_1576G OE_3246F	Q9HPL5 B0R5X2	[178]	14645280	
3c	HVO_0587	*cbiB*	yes	58%	VNG_1578H OE_3253F	Q9HPL3 B0R5X4	[178]	14645280	
3c	HVO_0592	*cbiZ*	yes	57%	VNG_1583C OE_3261F	Q9HPL3 B0R5X8	[179]	14990804	
3c	HVO_0589	*cobY*	yes	47%	VNG_1581C OE_3257F	Q9HPL1 B0R5X6	[180]	12486068	
3c	HVO_0588	*cobS*	yes	30%	STM2017	Q05602	[181]	17209023	
3c	-				STM0643	P39701	[182]	7929373	EC 3.1.3.73; CobC; no homolog in haloarchaea
3c	HVO_0586	*-*	prediction	-	-	-	[161]	12869542	EC 3.1.3.73; prediction for HSL01294 (VNG_1577C)
3c	pathway gap								EC 2.7.1.177
3c	HVO_0591	*cobD1*	yes	31%	STM0644	P97084	[183]	9446573	
3c	HVO_0593	*cobD2*	yes						no GSP; 51% seq_id to HVO_0591 (*cobD1*)
3c	HVO_0590	*cobT*	prediction				[161]	12869542	prediction for VNG_1572C
3c	halTADL_3045	*cobT*	yes	39%	STM0644	Q05603	[184]	8206834	
3d	HVO_B0051	*cobN*	yes	34%	-	P29929	[185]	1429466	
3d	HVO_B0051 (cont.)		no	29%	-	Q55284	[186] [187]	8663186 9716491	Mg chelatase
3d	HVO_B0050	*chlID*	no	46%	slr1030	P51634	[186] [187]	8663186 9716491	match to N-term; Mg chelatase
3d	HVO_B0050 (cont.)		no	33%	slr1777	P52772	[186] [187]	8663186 9716491	match to complete sequence, incl distant match to N-term; Mg chelatase
3e	HVO_2227	*ahbA*	yes	35%	-	I6UH61	[158]	21969545	
3e	HVO_2313	*ahbB*	yes	32%	-	I6UH61	[158]	21969545	
3f	HVO_1121	*ahbC*	yes	47%	Mbar_A1793	Q46BK8	[158] [188]	21969545 24669201	
3f	HVO_2144	*ahbD*			self		[162]	29284023	EC 1.3.98.6
3f	HVO_2144 (cont.)		yes	42%	Mbar_A1458	Q46CH7	[188]	24669201	
3f	HVO_1871	*chdC*			self		[162]	29284023	EC 1.3.98.5
3f	HVO_1871 (cont.)		yes	46%	BSU37670	P39645	[189]	28123057	

### 3.4. Coenzymes II: Coenzyme F420

Even though coenzyme F420 is predominantly associated with methanogenic archaea [190,191], it occurs also in bacteria, and a small amount of this coenzyme has been detected in non-methanogenic archaea, including halophiles [192]. The genes required for the biosynthesis of this coenzyme are encoded in haloarchaeal genomes, but the origin and attachment of the phospholactate moiety are not completely resolved (see below). To the best of our knowledge, only a single coenzyme F420-dependent enzymatic reaction has yet been reported for halophilic archaea [193]. Thus, the importance of this coenzyme in haloarchaeal biology is currently enigmatic and awaits experimental analysis.

(a) The pathway that creates the carbon backbone of this coenzyme has been reconstructed. We list the enzymes with their associated GSPs in Table 4. Coenzyme F420 contains a phospholactate moiety, which was reported to originate from 2-phospho-lactate [194], but this compound is metabolically not well-connected. As summarized in Appendix A, there are various new insights regarding this pathway from recent studies in other prokaryotes [195,196]. To the best of our knowledge, the haloarchaeal coenzyme F420 biosynthesis pathway has never been experimentally analyzed.

(b) The prediction of coenzyme F420-specific oxidoreductases in *Mycobacterium* and actinobacteria has been reported [197], leading to patterns and domains that are also found in haloarchaea. Several such enzymes are described in Appendix A.

(c) HVO_1937 might be a coenzyme F420-dependent 5,10-methylenetetrahydrofolate reductase (see, also, below: C1 metabolism, and Appendix A).

(d) The precursor for coenzyme F420 may be used by a photolyase involved in DNA repair.

**Table 4 genes-12-00963-t004:** Proteins with open annotation issues and their Gold Standard Protein homologs (Section 3.4). For a description of this table, see the legend to Table 1.

			Gold Standard Protein						
Section	Code	Gene	Isofunc	%seq_id	Locus Tag	UniProt	Reference	PMID	Comment
4a	HVO_2198	*cofH*	yes	35%	MJ1431	Q58826	[198] [199]	14593448 25781338	
4a	HVO_2201	*cofG*	yes	43%	MJ0446	Q57888	[198] [200] [199]	14593448 23072415 25781338	
4a	HVO_2202	*cofC*	yes	25%	MJ0887	Q58297	[194] [195] [196]	18260642 30952857 31469543	
4a	HVO_2479	*cofD*	yes	39%	MM_1874	Q8PVT6	[201] [196]	18252724 31469543	
4a	HVO_2479 (cont.)		yes	32%	MJ1256	Q58653	[202]	11888293	
4a	HVO_1936	*cofE*	yes	47%	AF_2256	O28028	[203]	17669425	
4a	HVO_1936 (cont.)		yes	38%	MJ0768	Q58178	[204]	12911320	
4b	HVO_0433	*npdG*	yes	38%	AF_0892	O29370	[205]	not in PubMed	
4b	HVO_B0113	-	no	27%	Rv0132c	P96809	[206]	24349169	too distant to assume isofunctionality
4b	HVO_B0342	-	unknown	29%	-	O93734	[207] [208]	8706724 15016352	too distant to assume isofunctionality
4b	NP_1902A	-	no	28%	-	Q9UXP0	[209] [210]	1735436 9933933	too distant to assume isofunctionality
4b	NP_4006A	-	no	27%	MJ0870	Q58280	[211]	16048999	too distant to assume isofunctionality
4c/5c	HVO_1937	*mer*	no	38%	MTH_1752	O27784	[212] [213] [214]	2298726 7649177 10891279	
4d	HVO_2911	*phr2*	yes	62%	VNG_1335G OE_2907R	Q9HQ46 B0R5D6	[215] [216]	2681164 12773185	
4d	HVO_2843	*phr1*	no	45%	sll1629	P77967	[217]	12535521	sll1629 implicated in transcription regulation
4d	HVO_2843 (cont.)		possibly	45%	At5g24850	Q84KJ5	[218] [219]	12834405 17062752	mediates photo-repair of ssDNA
4d	HVO_1234	*phr3*	possibly	40%	Atu4765	A9CH39	[220]	23589886	

### 3.5. Coenzymes III: Coenzymes of C1 Metabolism: Tetrahydrofolate in Haloarchaea and Methanopterin in Methanogens

Halophilic and methanogenic archaea use distinct coenzymes as one-carbon carriers (C1 metabolism): tetrahydrofolate in haloarchaea and methanopterin in methanogens [221,222]. Several characterized methanogenic proteins that act on or with methanopterin have comparably close homologs in haloarchaea (Table 5), which results in the misannotation of haloarchaeal proteins (e.g., in SwissProt) as being involved in methanopterin biology. We assume that the haloarchaeal proteins function with the haloarchaeal one-carbon carrier tetrahydrofolate and that this shift in coenzyme specificity is possible due to the structural similarity between methanopterin and tetrahydrofolate (a near-identical core structure consisting of a pterin heterocyclic ring linked via a methylene bridge to a phenyl ring) (Figure 2). A detailed review on the many variants of the tetrahydrofolate biosynthetic pathway is available [223].

(a) Folate biosynthesis requires aminobenzoate. We proposed candidates for a pathway from chorismate to para-aminobenzoate [66,224] (for details, see Appendix A). However, these predictions have not been adopted by KEGG (accessed April 2021), and without experimental confirmation, this is unlikely to ever happen.

(b) GTP cyclohydrolase MptA (HVO_2348) catalyzes a reaction in the common part of tetrahydrofolate and methanopterin biosynthesis. The enzymes specific for methanopterin biosynthesis are absent from haloarchaea, and thus, the assignment of HVO_2348 to the methanopterin biosynthesis pathway in UniProt is invalid (accessed March 2021). 

The next common pathway step (EC 3.1.4.56) has been resolved in *M. jannaschii* (MJ0837) but is still a pathway gap in halophilic archaea. MJ0837 is very distantly related to HVO_A0533, which is a promising candidate for experimental analysis.

HVO_2628 shows 30% protein sequence identity with the enzyme catalyzing the first committed step to methanopterin biosynthesis. As detailed in Appendix A, we consider it likely that it does not catalyze that reaction.

(c) Two enzymes that alter the oxidation level of the coenzyme-attached one-carbon compound probably function with tetrahydrofolate, even though their methanogenic homologs function with methanopterin. In contrast to their assignments in KEGG and UniProt (as of March 2021), their probable functions are thus methenyltetrahydrofolate cyclohydrolase (HVO_2573) and 5,10-methylenetetrahydrofolate reductase (HVO_1937) (see Figure 2 and Appendix A). 

**Table 5 genes-12-00963-t005:** Proteins with open annotation issues and their Gold Standard Protein homologs (Section 3.5). For a description of this table, see the legend to Table 1.

			Gold Standard Protein						
Section	Code	Gene	Isofunc	%seq_id	Locus Tag	UniProt	Reference	PMID	Comment
5a	HVO_0709	*pabA*	no	47%	TTHA1843	P05379	[225]	2844259	Trp biosynthesis
5a	HVO_0709 (cont.)		yes/no	39%	BSU00750	P28819	[226]	2123867	TrpG works with TrpE and with PabB
5a	HVO_0710	*pabB*	no	46%	TTHA1844	P05378	[225]	2844259	Trp biosynthesis
5a	HVO_0710 (cont.)		yes	44%	BSU00740	P28820	[227]	19275258	PabB; para-aminobenzoate biosynthesis
5a	HVO_0708	*pabC*	no	36%	AF_0933	O29329	[228]	30733943	branched-chain amino acids
5b	HVO_2348	*mptA*			self		[229]	19478918	gene deletion phenotypes
5b	HVO_2348 (cont.)		yes	41%	MJ0775	Q58185	[230]	17497938	common part of methanopterin and tetrahydrofolate biosynthesis
5b	HVO_A0533	-	unknown	27%	MJ0837	Q58247	[231]	19746965	if isofunctional would resolve a pathway gap
5b	HVO_2628	-	no	31%	AF_2089	O28190	[232]	12142414	first committed step to methanopterin biosynthesis
5b	HVO_2628 (cont.)		no	26%	MJ1427	Q58822	[233]	15262968	first committed step to methanopterin biosynthesis
5c	HVO_2573	*mch*	no	45%	MK0625	P94954	[234]	9676239	acts on a one-carbon attached to methanopterin
4c/5c	HVO_1937	*mer*	no	38%	MTH_1752	O27784	[212] [213] [214]	2298726 7649177 10891279	acts on a one-carbon compound attached to methanopterin

### 3.6. Coenzymes IV: NAD and FAD (Riboflavin)

(a) The energy source for NAD kinase may be ATP or polyphosphate. This is unresolved for the two paralogs of probable NAD kinase (HVO_2363, *nadK1* and HVO_0837, *nadK2*). These show only 25% protein sequence identity to each other (see Appendix A). Polyphosphate was not found in exponentially growing *Hfx. volcanii* cells [235], and thus ATP is the more likely energy source.

(b) HVO_0781 is encoded in nearly all haloarchaeal genomes, according to OrthoDB, and shows very strong syntenic coupling with the adjacent gene, HVO_0782, according to SyntTax analysis. Characterized homologs to HVO_0781 cleave S-adenosyl-methionine into methionine and adenosine, a reaction that seems wasteful. If so, then this gene would not be expected to be retained in most species and neither would it maintain a strongly conserved gene clustering (see Appendix A). HVO_0782 is an enzyme involved in NAD biosynthesis, which is encoded in most haloarchaeal and archaeal genomes. Thus, HVO_0781 is also a candidate for being involved in NAD biosynthesis. 

(c) We described the reconstruction of riboflavin biosynthesis based on a detailed bioinformatic reconstruction [236]. The enzymes and their associated GSPs are listed in Table 6. Three pathway gaps remain, with candidate genes predicted for two of these [236] (for details, see Appendix A).

**Table 6 genes-12-00963-t006:** Proteins with open annotation issues and their Gold Standard Protein homologs (Section 3.6). For a description of this table, see the legend to Table 1.

			Gold Standard Protein						
Section	Code	Gene	Isofunc	%seq_id	Locus Tag	UniProt	Reference	PMID	Comment
6a	HVO_2363	*nadK1*	unclear	37%	Rv1695	P9WHV7	[237]	11006082	can use ATP and PP
6a	HVO_2363 (cont.)		unclear	31%	AF_2373	O30297			ATP or PP usage unresolved
6a	HVO_0837	*nadK2*	unclear	28%	Rv1695	P9WHV7			can use ATP and PP
6a	HVO_0837 (cont.)		unclear	partial	AF_2373	O30297			ATP or PP usage unresolved
6b	HVO_0782	*nadM*	yes	53%	MJ0541	Q57961	[238] [239]	9401030 10331644	
6b	HVO_0781	*-*	unknown	42%	Sare_1364	A8M783	[240]	18720493	
6b	HVO_0781 (cont.)		unknown	35%	PH0463	O58212	[241]	18551689	
6c	HVO_0327	*ribB*	yes	43%	MJ0055	Q60364	[242]	12200440	
6c	HVO_0974	*ribH*	yes	45%	MJ0303	Q57751	[243]	12603336	
6c	HVO_1284	*arfA*		self			[244]	21999246	gene deletion leads to riboflavin auxotrophy
6c	HVO_1284 (cont.)		yes	44%	MJ0145	Q57609	[245]	12475257	
6c	HVO_1235	-	prediction				[236]	28073944	*arfB* candidate
6c	HVO_1341	*arfC*	yes	36%	MJ0671	Q58085	[246] [247]	11889103 18671734	
6c	HVO_2483	*-*	prediction	34%	MJ0699	Q58110	[236]	28073944	also predicted for MJ0699
6c	pathway gap								EC 3.1.3.104
6c	HVO_0326	*rbkR*	yes	37%	TA1064	Q9HJA6	[236]	28073944	bifunctional as gene regulator and enzyme
6c	HVO_0326 (cont.)		yes/no	32%	MJ0056	Q60365	[248]	18073108	enzyme only; lacks an N-terminal HTH domain
6c	HVO_1015	*ribL*	yes	50%	MJ1179	Q58579	[249]	20822113	

### 3.7. Biosynthesis of Membrane Lipids, Bacterioruberin and Menaquinone

Archaeal membrane lipids contain ether-linked isoprenoid side chains (see [250] and the references cited therein). The isoprenoid precursor isopentenyl diphosphate is synthesized in haloarchaea by a modified version of the mevalonate pathway [251]. Isoprenoid units are then linearly condensed into the C20 compound geranylgeranyl diphosphate. The haloarchaeal core lipid, archaeol, consists of 2,3-sn-glycerol with two C20 isoprenoid side chains attached by ether linkages. In some archaea, especially alkaliphiles, C25 isoprenoids are also found (see, e.g., [252,253]). Additionally, a number of distinct headgroups are found in polar lipids (phospholipids) (reviewed in [250]) (Figure 3). Even though polar lipids are used as important taxonomic markers [254], their biosynthetic pathways are not completely resolved.

Haloarchaea typically have a red color, which is due to carotenoids, mainly the C50 carotenoid bacterioruberin [255,256,257]. For carotenoid biosynthesis, two molecules of geranylgeranyl diphosphate, a C20 compound, are linked head-to-head to generate phytoene, which is desaturated to lycopene [66,258]. The pathway from lycopene to the C50 compound bacterioruberin has been experimentally characterized [257,259].

(a) We assigned HVO_2725 (*idsA1*, paralog of NP_3696A) and HVO_0303 (*idsA2*, paralog of NP_0604A) for the linear isoprenoid condensation reactions, resulting in a C20 isoprenoid (EC 2.5.1.10 and EC 2.5.1.29, short-chain isoprenyl diphosphate synthase) (see, also, Appendix A). Some archaea, mainly haloalkaliphiles, also contain C25 isoprenoid side chains. Geranylfarnesyl diphosphate synthase, the enzyme that generates the C25 isoprenoids, has been purified and enzymatically characterized from *Nmn. pharaonis* [260], but data required for the assignment to a specific gene have not been collected. Three paralogous genes from *Nmn. pharaonis* are candidates for this function (NP_0604A, NP_3696A and NP_4556A). Since NP_0604A and NP_3696A have orthologs in *Hfx. volcanii*, a species devoid of C25 lipids, we assigned the synthesis of C25 isoprenoids (geranylfarnesyl diphosphate synthase activity) to the third paralog, NP_4556A. UniProt assigned C25 biosynthesis activity to NP_3696A for undescribed reasons (as of April 2021), and KEGG does not make this assignment for any of the three paralogs (as of April 2021). Our assignments are supported by an analysis of the key residues that determine the length of the isoprenoid chain [261]. These authors labeled the cluster containing NP_3696A (WP011323557.1) as “C15/C20” and the cluster containing NP_4556A (WP011323984.1) as “C20->C25->C30?”.

(b) Typical polar lipids in haloarchaea (Figure 3) are phosphatidylglycerophosphate methyl ester (PGP-Me) and phosphatidylglycerol (PG) but, also, phosphatidylglycerosulfate (PGS) [261,262,263]. Other polar lipids are archaetidylserine and its decarboxylation product archaetidylethanolamine, both of which are found in rather low quantities in *Haloferax* [264]. A third group of polar lipids has a headgroup derived from myo-inositol. The biosynthetic pathway of the headgroup is only partially resolved. One CDP-archaeol 1-archaetidyltransferase that belongs to a highly conserved three-gene operon may attach either glycerol phosphate or myo-inositol phosphate. In Appendix A, we summarize the arguments in favor of each of these candidates, but the true function can only be decided by experimental analysis.

(c) Carotenoid biosynthesis involves the head-to-head condensation of the C20 isoprenoid geranylgeranyl diphosphate to phytoene, which is desaturated to lycopene [66,258]. The *crtB* gene product (e.g., HVO_2524) catalyzes the head-to-head condensation. It is yet uncertain which gene product is responsible for the desaturation of phytoene to lycopene. The further pathway from lycopene to bacterioruberin has been experimentally characterized in *Haloarcula japonica* [257]. A three-gene cluster (*crtD*-*lyeJ*-*cruF*) codes for the three enzymes of this pathway. The synteny of this three gene cluster is strongly conserved, according to SyntTax analysis. Several genes that are certainly or possibly involved in carotenoid biosynthesis are encoded in the vicinity of this cluster (for details, see Appendix A).

(d) Halophilic archaea contain menaquinone as a lipid-based two-electron carrier of the respiratory chain [264,265]. We described the reconstruction of the menaquinone biosynthesis pathway (Table 7), with two pathway gaps remaining open (see Appendix A for details).

**Table 7 genes-12-00963-t007:** Proteins with open annotation issues and their Gold Standard Protein homologs (Section 3.7). For a description of this table, see the legend to Table 1.

			Gold Standard Protein						
Section	Code	Gene	Isofunc	%seq_id	Locus Tag	UniProt	Reference	PMID	Comment
7a	NP_0604A	*idsA2*	yes	32%	GACE_1337	A0A0A7GEY4	[266]	30062607	ortholog of HVO_0303 (66%); produces a C20 isoprenoid (same assignment for NP_0604A)
7a	NP_0604A (cont.)	*idsA2*	no	30%	APE_1764	Q9YB31 Q9UWR6	[267]	10632701	produces a C25 isoprenoid (C20 assigned to NP_0604A)
7a	NP_3996A	*idsA3*	yes	44%	GACE_1337	A0A0A7GEY4	[266]	30062607	ortholog of HVO_2725 (67%); produces a C20 isoprenoid (same assignment for NP_3996A)
7a	NP_3996A (cont.)	*idsA2*	no	36%	APE_1764	Q9YB31 Q9UWR6	[267]	10632701	produces a C25 isoprenoid (C20 assigned to NP_3996A)
7a	NP_4556A	*idsA1*	no	34%	GACE_1337	A0A0A7GEY4	[266]	30062607	no ortholog in *Hfx. volcanii*; produces a C20 isoprenoid (C25 assigned to NP_4556A)
7a	NP_4556A (cont.)	*idsA1*	yes	29%	APE_1764	Q9YB31 Q9UWR6	[267]	10632701	produces a C25 isoprenoid (same assignment for NP_4556A)
7b	HVO_0332	*carS*	yes	45%	AF_1740	O28537	[268]	25219966	
7b	HVO_1143	*assA*	yes	32%	MTH_1027	O27106	[269]	12562787	gene synonym: *pgsA3*
7b	HVO_1297	*aisA*	yes	25%	MTH_1691	O27726	[270]	19740749	gene synonym: *pgsA2*
7b	HVO_1136	*pgsA1*	-						only distant partial matches to GSPs
7b	HVO_1971	*pgsA4*	unclear	26%	MTH_1027	O27106	[269]	12562787	MTH_1027 is less distant to HVO_1143
7b	HVO_0146	*asd*	no	39%	SMc00551	Q9FDI9	[271]	18708506	equivalent function for the bacterial lipid
7b	HVO_1295	*hisC*		self			[272]	2345144	complements a His auxotrophy mutant
7b	HVO_1295 (cont.)		yes	31%	b2021	P06986	[273]	2999081	weak support, see text
7b	HVO_1296	*adk2*	unclear	34%	PAB0757	Q9UZK4	[274]	24823650	*Pyrococcus*: involved in ribosome biogenesis
7b	HVO_1296 (cont.)		unclear	32%	-	Q9Y3D8	[275]	15630091	human: adenylate kinase; HVO_1296 may be inositol kinase
7b	HVO_2496	*adk1*	yes	45%	BSU01370	P16304	[276]	31111079	*Bacillus*: adenylate kinase
7b	HVO_B0213	-	yes	43%	AF_1794	O28480	[277] [278]	11015222 22261071	*Archaeoglobus*: adenylate kinase
7b	HVO_1135	*-*	-						a SAM-dependent methyltransferase
7c	HVO_2524	*crtB*		self			[9] [279]	25488358 29038254	*crtB* mutants are colorless
7c	HVO_2524 (cont.)		yes	32%	Synpcc7942 _1984	P37269	[280]	1537409	
7c	HVO_2527	*lyeJ*		self			[259]	21840984	
7c	HVO_2527 (cont.)		yes	65%	VNG_1682C OE_3380R	Q9HPD9 B0R651	[259]	21840984	
7c	HVO_2527 (cont.)		yes	61%	C444_12922	M0L7V9	[257]	25712483	
7c	HVO_2528	*crtD*		self			[279]	29038254	a HVO_2528 mutant was white
7c	HVO_2528 (cont.)		yes	71%	C444_12917	A0A0A1GKA2	[257]	25712483	
7c	HVO_2526	*cruF*	yes	59%	C444_12927	A0A0A1GNF2	[257]	25712483	
7d	HVO_1470	*menF*	yes	38%	PA4231	Q51508	[281]	7500944	
7d	HVO_1469	*menD*	yes	37%	BSU30820	P23970	[282]	20600129	
7d	pathway gap								EC 4.2.99.20
7d	HVO_1461	*menC*	no	29%	BSU12980	O34508	[283]	11747447	Ala/Glu epimerase
7d	HVO_1461 (cont.)		yes	24%	BSU30780	O34514	[284]	10194342	o-succinylbenzoate synthase
7d	HVO_1375	*menE*	yes	36%	BSU30790	P23971	[285]	27933791	
7d	HVO_1465	*menB*	yes	66%	Rv0548c	P9WNP5	[286]	20643650	
7d	pathway gap								EC 3.1.2.28
7d	HVO_1462	*menA*	yes	37%	b3930	P32166	[287]	9573170	
7d	HVO_0309	*menG*	yes/no	44%	At3g63410	Q9LY74	[288]	14508009	*A. thaliana* enzyme also involved in tocopherol biosynthesis
7d	HVO_0309 (cont.)		yes	27%	-	O86169	[289]	9139683	

### 3.8. Issues Concerning RNA Polymerase, Protein Translation Components and Signal Peptide Degradation

(a) Haloarchaeal RNA polymerase consists of a set of canonical subunits (encoded by *rpoA1A2B1B2DEFHKLNP*). *Hbt. salinarum* and a subset of other haloarchaea contain an additional subunit called epsilon [290,291]. Purified RNA polymerase containing the epsilon subunit transcribes native templates efficiently, in contrast to the RNA polymerase devoid of this subunit [291]. The biological relevance of this subunit is enigmatic (see Appendix A).

(b) Two distant paralogs are found for haloarchaeal ribosomal protein S10 (uS10) in nearly all haloarchaeal genomes. It is uncertain if both occur in the ribosome, whether they occur together or are mutually exclusive. The latter distribution would result in heterogeneity of the ribosomes. Alternatively, one of the paralogs may exclusively have a non-ribosomal function. 

In a subset of archaea, two distant paralogs are found for haloarchaeal ribosomal protein S14 (uS14) (ca 20% of the genomes, e.g., in *Nmn. pharaonis*). For more details, see Appendix A.

(c) The ribosomal protein L43e (eL43) shows heterogeneity with respect to the presence of the C2–C2-type zinc finger motif. This zinc finger is found in L43e from all *Halobacteriales* and all euryarchaeal proteins outside the order *Halobacteria* but is not found in *Haloferacales* and is very rare in *Natrialbales*. Eukaryotic orthologs (e.g., from rat and yeast) contain this zinc finger, and its biological importance has been experimentally shown for the yeast protein [292] (for details, see Appendix A).

(d) Diphthamide is a complex covalent modification of a histidine residue of translation elongation factor a-EF2. This pathway has been reconstructed (Table 8) based on distant homologs (enzymes encoded by *dph2* and *dph5*) and by a detailed bioinformatic analysis (enzyme encoded by *dph6*) [293] (for details, see Appendix A). These uncertain function assignments await experimental confirmation.

(e) N-terminal signal sequences target proteins to the secretion machinery. Subsequent to membrane insertion or transmembrane transfer, the signal sequence is cleaved off by a signal peptidase. After cleavage, the signal peptide must be degraded to avoid clogging of the membrane. Degradation is catalyzed by signal peptide peptidase. Candidates for this activity have been predicted from two protein families [294,295] (for details, see Appendix A).

**Table 8 genes-12-00963-t008:** Proteins with open annotation issues and their Gold Standard Protein homologs (Section 3.8). For a description of this table, see the legend to Table 1.

			Gold Standard Protein						
Section	Code	Gene	Isofunc	%seq_id	Locus Tag	UniProt	Reference	PMID	Comment
8a	OE_1279R	*rpoeps*		self			[290] [291]	2495365 6852054	
8b	HVO_0360	*rps10a*	yes	94%	rrnAC2405	P23357	[296]	1764513	
8b	HVO_1392	*rps10b*	-						no GSP; 24% seq_id to HVO_0360 (*rps10a*)
8b	NP_4882A	*rps14a*	yes	72%	rrnAC1597.1	P26816	[297]	1832208	full-length similarity; *Haloarcula* protein was not isolated or characterized
8b	NP_4882A (cont.)		yes	57%	YDL061C	P41058	[298]	18782943	yeast YS29B; N-term 20 aa divergent
8b	NP_1768A	*rps14b*	unclear	80%	rrnAC1597.1	P26816	[297]	1832208	N-term 20 aa divergent
8c	OE_1373R	*rpl43e*	yes	69%	rrnAC1669	P60619	[299]	10937989	
8c	OE_1373R (cont.)		yes	39%	YPR043W	P0CX25	[292] [300]	10588896 11866512	
8c	HVO_0654	*rpl43e*	yes	54%	rrnAC1669	P60619	[299]	10937989	*Haloarcula*: has zinc finger; *Haloferax*; lacks zinc finger
8d	HVO_1631	*dph2*	yes	35%	PH1105	O58832	[301]	20931132	
8d	HVO_0916	*dph5*	yes	39%	PH0725	O58456	[302]	20873788	
8d	HVO_1077	*dph6*	yes	31%	YLR143W	Q12429	[303] [304]	23169644 23468660	
8e	HVO_0881	*sppA1*	yes	33%	BSU19530	O34525	[305] [306]	10455123 22472423	
8e	HVO_1987	*sppA2*	probably	23%	BSU19530	O34525	[305] [306]	10455123 22472423	
8e	HVO_1107	*-*	prediction						no GSP

### 3.9. Miscellaneous Metabolic Enzymes and Proteins with Other Functions

Here, we list a few other enzymatic or nonenzymatic functions for which candidate genes have been assigned but without experimental validation.

(a) Ketohexokinase from *Haloarcula vallismortis* has been experimentally characterized [307]. However, the activity was not assigned to a gene. Detailed bioinformatic analyses have been made [308,309] and point to a small set of orthologs represented by Hmuk_2662, the ortholog of HVO_1812 (for further details, see Appendix A).

(b) The assignment of fructokinase activity to the *Hht. litchfieldiae* candidate gene halTADL_1913 (UniProt:A0A1H6QYL4) is based on a differential proteomic analysis [309] (see Appendix A for details). Very close homologs are rare in haloarchaea. For this protein family (carbohydrate kinase), it is unclear if more distant homologs (with about 50% protein sequence identity) are isofunctional.

(c) A candidate gene for glucoamylase is HVO_1711 for the reasons described in Appendix A. The enzyme from *Halorubrum sodomense* has been characterized [310], but the activity has not yet been assigned to a gene.

(d) A strong candidate for having glucose-6-phosphate isomerase activity is *Hfx. volcanii* HVO_1967 (*pgi*), based on 36% protein sequence identity to the characterized enzyme from *M. jannaschii* (MJ1605) [311] (Table 9).

**Table 9 genes-12-00963-t009:** Proteins with open annotation issues and their Gold Standard Protein homologs (Section 3.9). For a description of this table, see the legend to Table 1.

			Gold Standard Protein						
Section	Code	Gene	Isofunc	%seq_id	Locus Tag	UniProt	Reference	PMID	Comment
9a	HVO_1812	*-*	prediction						no GSP
9b	halTADL_1913	*-*	yes	37%	-	P26984	[312]	1809835	
9b	halTADL_1913 (cont.)	*-*	yes	31%	OCC_03567	Q7LYW8 H3ZP68	[313]	15138858	
9c	HVO_1711	*-*	probably	33%	-	P29761	[314]	1633799	P29761 matches to C-term half of HVO_1711
9c	HVO_1711 (cont.)	*-*	probably	51%	SAMN 04487937_ 2677	A0A1I6HD35	[310]	8305855	correlation between PMID:8305855 and A0A1I6HD35 likely (see text)
9d	HVO_1967	*pgi*	yes	36%	MJ1605	Q59000	[311]	14655001	
9e	OE_1665R	*kdgA*	no	31%	PA1010	Q9I4W3	[139]	21396954	GSP for *dapA* (see under 2a)
9e	OE_1665R (cont.)		probably	30%	TTX_1156.1 TTX_1156a	G4RJQ2	[315]	15869466	
9e	OE_1665R (cont.)		probably	25%	SSO3197	Q97U28	[315]	15869466	
9f	HVO_1692	*ludB*		self			[21]	30707467	
9f	HVO_1692 (cont.)		probably	35%	BSU34040	O07021	[316]	19201793	matches up to HVO_1692 pos 490 of 733
9f	HVO_1692 (cont.)		probably	35%	PST_3338	O4VPR6	[317]	25917905	matches up to HVO_1692 pos 400 of 733
9f	HVO_1693	*ludC*		self			[21]	30707467	
9f	HVO_1693 (cont.)		probably	30%	BSU34030	O32259	[316]	19201793	
9f	HVO_1693 (cont.)		probably	33%	PST_3339	O4VPR7	[317]	25917905	partial match
9f	HVO_1697	*-*	unclear	24%	PST_3340	O4VPR8	[317]	25917905	
9f	HVO_1696	*lctP*	probably	44%	PST_3336	O4VPR4	[317]	25917905	
9g	HVO_B0300	*pucL1*	yes	49%	BSU32450	O32141	[318]	20168977	*Bacillus*: bifunctional, matches to C-term
9g	HVO_B0299	*pucM*	yes	43%	BSU32460	O32142	[319]	16098976	
9g	HVO_B0301	*pucL2*	yes	43%	BSU32450	O32141	[320]	17567580	*Bacillus*: bifunctional, matches to N-term
9g	HVO_B0302	*pucH1*	no	33%	-	Q8VTT5	[321]	12148274	paper in Chinese, abstract in English; pyrimidine degradation
9g	HVO_B0302 (cont.)		yes	30%	STM0523	Q7CR08	[322]	23287969	purine degradation
9g	HVO_B0302 (cont.)		yes	29%	BSU32410	O32137	[323]	11344136	purine degradation
9g	HVO_B0306	*amaB4*	no	39%	-	Q53389	[324]	22904279	carbamoyl-AA hydrolysis
9g	HVO_B0306 (cont.)		yes	34%	At5g43600	Q8VXY9	[325] [326]	19935661 23940254	purine degradation
9g	HVO_B0308	*coxS*	no	46%	Saci_2270	Q4J6M5	[327]	10095793	GAPDH
9g	HVO_B0308 (cont.)		no	41%	-	P19915	[328]	10482497	CO-DH
9g	HVO_B0308 (cont.)		yes	39%	b2868	Q46801	[329]	10986234	xanthine DH
9g	HVO_B0309	*coxL*	yes	33%	b2866	Q46799	[329]	10986234	xanthine DH
9g	HVO_B0309 (cont.)		no	28%	-	P19913	[328]	10482497	CO-DH
9g	HVO_B0309 (cont.)		no	26%	Saci_2271	Q4J6M3	[327]	10095793	GAPDH
9g	HVO_B0310	*coxM*	no	31%	Saci_2269	Q4J6M6	[327]	10095793	GAPDH
9g	HVO_B0310 (cont.)		no	31%	-	P19914	[328]	10482497	CO-DH
9g	HVO_B0310 (cont.)		yes	25%	b2867	Q46800	[329]	10986234	xanthine DH
9g	HVO_B0303	*uraA4*	yes	38%	b3654	P0AGM9	[330]	16096267	
9h	HVO_0197	*-*	possibly	39%	lp_0105	F9UST0	[331]	27114550	LarB family protein
9h	HVO_2381	*-*	possibly	31%	lp_0106/ lp_0107	F9UST1	[331]	27114550	LarC family protein
9h	HVO_0190	*-*	possibly	34%	lp_0109	F9UST4	[331]	27114550	LarE family protein
9i	HVO_1660	*dacZ*		self			[37]	30884174	
9i	HVO_0756	*-*	prediction				[332]	32095817	
9i	HVO_0990	*-*	prediction				[332]	32095817	
9i	HVO_1690	*-*	prediction				[332]	32095817	
9j	HVO_2763	*-*		self			[333]	22350204	no function could be assigned
9j	HVO_2763 (cont.)		no	27%	HVO_0144	D4GZ88	[334]	18437358	Rnase Z
9k	HVO_2410	*dabA*	yes	33%	Hneap_0211	D0KWS7	[335]	31406332	
9k	HVO_2411	*dabB*	yes	31%	Hneap_0212	D0KWS8	[335]	31406332	

(e) A candidate gene for specifying an enzyme with 2-dehydro-3-deoxy-(phospho)gluconate aldolase activity is *Hbt. salinarum kdgA* (OE_1665R). It is rather closely related (36% protein sequence) to *Hfx. volcanii* HVO_1101 (encoded by *dapA*), which is involved in lysine biosynthesis, a biosynthetic pathway that is absent from *Hbt. salinarum*. The function assignment is based on distant homologs from *Saccharolobus (Sulfolobus) solfataricus* and *Thermoproteus tenax*, which have been characterized [315] (for details, see Appendix A).

(f) Haloarchaea may contain an NAD-independent L-lactate dehydrogenase, LudBC (HVO_1692 and HVO_1693). The deletion of this gene pair impairs growth on rhamnose, which is catabolized to pyruvate and lactate [21]. There is a very distant relationship (for details, see Appendix A) to the LldABC subunits of the characterized L-lactate dehydrogenase from *Pseudomonas stutzeri* A1501 [317] and to the LutABC proteins from *B. subtilis*, which have been shown to be involved in lactate utilization [316].

(g) *Hfx. volcanii* may be able to convert urate into allantoin using the gene cluster HVO_B0299-HVO_B0302. This could be part of a complete degradation pathway for purines, but this has to be considered highly speculative (see Appendix A).

(h) *Hfx. volcanii* may contain an enzyme having a “nickel-pincer cofactor”. The biogenesis of this cofactor may be catalyzed by *larBCE* (as detailed in Appendix A). 

(i) Cyclic di-AMP (c-di-AMP) is an important nucleotide signaling molecule in bacteria and archaea. It is generated from two molecules of ATP by diadenylate cyclase (encoded by *dacZ*) and is degraded to pApA by phosphodiesterases [336]. The level of this signaling molecule is strictly controlled [337,338], thus requiring a sophisticated interplay of cyclase and phosphodiesterase. DacZ from *Hfx. volcanii* has been characterized, and it was shown that the c-di-AMP levels must be tightly regulated [37]. The degrading enzyme, however, has not yet been identified in *Haloferax*, but candidates have been proposed [332,336,339] (see Appendix A). 

(j) HVO_2763 is distantly related to RNase Z (HVO_0144, *rnz*). The experimental characterization of HVO_2763 [333] excluded activity as an exonuclease but did not reveal its physiological function. Upon transcriptome analysis, the downregulation of several genes was detected. Several of these were uncharacterized at the time of the experiment but have since been shown to be involved in the minor N-glycosylation pathway that was initially detected under low-salt conditions (see Appendix A for further details).

(k) A pair of genes (*dabAB*, HVO_2410 and HVO_2411) is predicted to function as a carbon dioxide transporter, based on the identification of such transporters in *Halothiobacillus neapolitanus* [335]. Being a member of the proton-conducting membrane transporter family, this protein may be misannotated as a subunit of the *nuo* or *mrp* complexes (see Appendix A for further details).

## 4. Conclusions

We described a large number of cases where the protein function cannot be correctly predicted when restricting considerations to the computational analyses without taking the biological contexts into account. An example was the switch from methanopterin to tetrahydrofolate as a C1 carrier in haloarchaea. Homologous enzymes, inherited from the common ancestor, have adapted to the new C1 carrier, rather than being replaced by non-homologous proteins. Function prediction tools may misannotate haloarchaeal proteins to work with methanopterin. Another example was the *nuo* complex and its misannotation as a type I NADH dehydrogenase. In other cases, even a distant sequence similarity may allow a valid function prediction if additional evidence (e.g., from a gene neighborhood analysis or from a detailed evaluation of the metabolic pathway gaps) is taken into account. Examples include cobalamin cluster proteins, which probably close the two residual pathway gaps, and the predicted degradation pathway for purines. In all these cases, we presented reasonable hypotheses based on the current knowledge, and in many cases, these were so well-supported as to be compelling, but to be certain, experimental data are required. With this overview, we attempted to arouse the curiosity of our colleagues, hoping that they will confirm or disprove our speculations and, thus, advance the knowledge about haloarchaeal biology. *Hfx. volcanii* is a model species for halophilic archaea, and the more complete and correctly its genome is annotated, the higher will be its value for system biology analyses (modeling) and for synthetic biology (metabolic engineering) and biotechnology.

## Figures and Tables

**Figure 1 genes-12-00963-f001:**
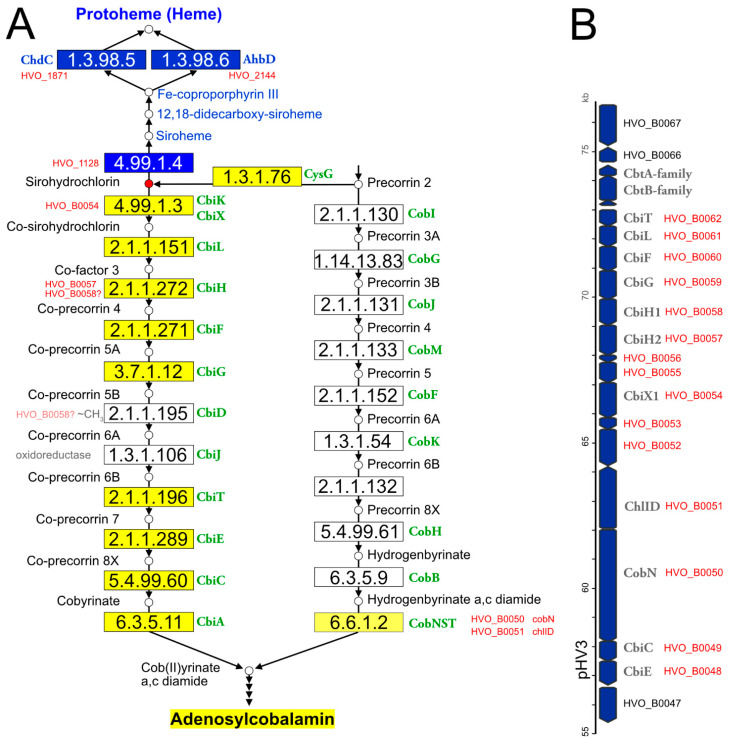
Illustration of the haloarchaeal cobalamin and heme biosynthesis pathways and of the major cobalamin biosynthesis gene cluster. (**A**) Biosynthesis pathways. This illustration is based on the corresponding KEGG map 00860. Small circles represent pathway intermediates and have their names assigned. Pathway intermediates upstream of precorrin-2 are not displayed. The circle for sirohydrochlorin is highlighted in red, as this is the branchpoint for heme and cobalamin biosynthesis in haloarchaea. Enzymatic reactions are shown by arrows, the EC numbers being provided in rectangular boxes. Rectangles are colored when the enzyme has been reconstructed for haloarchaea (blue: heme biosynthesis; dark yellow: de novo cobalamin biosynthesis; light yellow: late cobaltochelatase, which may be a salvage reaction). Gene names in green are adopted from KEGG and represent those from bacterial model pathways. Consecutive arrowheads indicate reaction series that are not shown in detail for space reasons. Additionally, some enzymes of the heme biosynthesis pathway are omitted for space reasons. For enzymatic reactions that are considered to be open issues, *Hfx. volcanii* locus tags are provided. For two pathway gaps (white boxes in the cobalt-early pathway), the type of reaction is indicated (oxidoreductase and ~CH3, indicating a methylation reaction). The question mark after HVO_B0058 indicates that this protein, currently co-attributed to EC 2.1.1.272, is a candidate for the yet-unassigned EC 2.1.1.195 reaction. We note that haloarchaea might use a deviating biosynthesis pathway, e.g., by swapping the methylation and oxidoreductase reactions (not illustrated). (**B**) The major cobalamin cluster, encoded on megaplasmid pHV3. Arrows are used to indicate the coding strand and are roughly drawn to scale. If assigned, the gene name is provided in addition to the *Hfx. volcanii* locus tag. Locus tags in red indicate genes that are part of the cobalamin cluster.

**Figure 2 genes-12-00963-f002:**
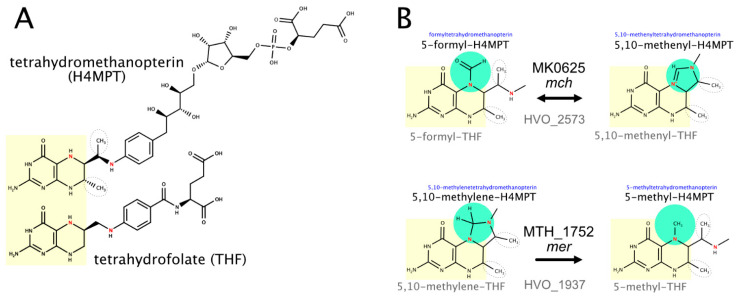
The structure of the C1 coenzymes tetrahydrofolate and methanopterin and two enzymes that act on the attached C1 compound. (**A**) The structures of tetrahydromethanopterin (top) and tetrahydrofolate (bottom) illustrate the similarities and differences between these C1 coenzymes. The common pteridine-based ring system is highlighted in yellow, and the initial biosynthesis step that generates this ring system is catalyzed by homologous enzymes (topic (b)). Two methanopterin-specific methyl groups are outlined by dashed ovals. N5 and N10, which are involved in the binding of the C1 compound, are colored red. (**B**) Two enzymatic reactions that alter the oxidation level of the C1 compound are illustrated. The methanogenic and haloarchaeal enzymes are homologous, even though they use distinct C1 coenzymes (topic (c)). It should be noted that MTH-1752 uses coenzyme F420 (not illustrated, Section 3.4, topic (c)), and this might also hold true for HVO_1937.

**Figure 3 genes-12-00963-f003:**
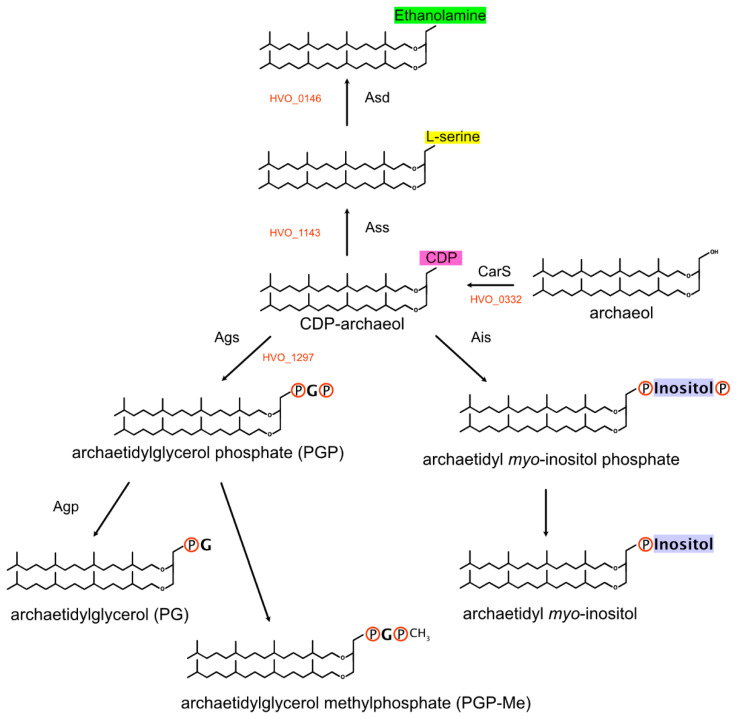
Biosynthesis of polar lipids. A key intermediate is CDP-archaeol, which is generated from archaeol (displayed as fully saturated) by CarS. Members of the InterPro:IPR000462 family then transfer the CDP-archaeol to the hydroxyl group (alcohol group) of the target molecule (backbone: serine, glycerol and myo-inositol). Subsequent modifications contribute to the diversity of polar lipids.

**Table 1 genes-12-00963-t001:** Proteins with open annotation issues and their Gold Standard Protein homologs (Section 3.1).

			Gold Standard Protein						
Section	Code	Gene	Isofunc	%seq_id	Locus Tag	UniProt	Reference	PMID	Comment
1a	HVO_1305 HVO_1304	*porAB*	yes	67% 80%	OE2623R OE2622R	B0R4 × 6 B0R4 × 5	[87] [88] [89]	1555599 6266826 6266827	
1a	HVO_0888 HVO_0887	*korAB*	yes	77% 77%	OE1711R OE1710R	B0R3G0 B0R3F9	[88] [89]	6266826 6266827	
1a/1b	HVO_2995	*fdx*	yes	88%	OE4217R	B0R7I9	[100] [101] [86]	964365 188650 201489	role in oxidative decarboxylation
1a/1b	HVO_2995 (cont.)				self	D4GY89	[90]	22103537	role in nitrate assimilation
1c	HVO_0979 (complex)	*nuoB*	possibly	50%	tlr0705	Q8DKZ4	[102] [103] [104]	15910282 30573545 32001694	reoxidizes ferredoxin
1c	HVO_0979 (cont.)		no	48%	b2287	P0AFC7	[92] [93]	7607227 9485311	reoxidizes NADH in *E.coli*
1d	NP_3508A	*ndh1*	special	26% (N-term 140 aa)	-	Q7ZAG8			function of Q7ZAG8 was reassigned (from ndh1 to sqr) after annotation transfer
1d	NP_3508A (cont.)		possibly	30%	BpOF4_04810	A7LKG4	[105]	18359284	type II NADH dehydrogenase
1e	HVO_2620 HVO_0842 HVO_0841	*petABD*	yes	39%	SYNPCC7002_ A0842	P28056	[106]	11245788	HVO_0842 (*petB*) related to cytochrome b6
1f	HVO_2810	*sdhD*	yes	66%	NP_4268A	Q3INS7	[81] [94]	9109654 PhD_Mattar	
1g	HVO_0943	*cbaD*	yes	57%	NP_2966A	A0A1U7EWW4	[107]	9428682	
	HVO_0943 (cont.)		-	63%	OE_4073R (C-term)	B0R7A9		-	halocyanin/cbaD fusion protein, uncharacterized
1g	HVO_2150	*hcpG*	-	44%	OE_4073R (N-term)	B0R7A9		-	halocyanin/cbaD fusion protein, uncharacterized
1h	HVO_0945 (complex)	*cbaA*	yes	64%	NP_2966A	A0A1U7EWW4	[107]	9428682	
1h	HVO_0907 (complex)	*coxA1*			self		[108]	11790755	
1h	HVO_0907 (cont.)		yes	70%	VNG_0657G (OE_1979R)	P33588	[109] [110]	2542239 1659810	
1h	HVO_1645 (complex)	*coxAC2*	yes	43%	APE_0793.1	Q9YdX6	[111]	12471503	
1h	HVO_0462 HVO_0461	*cydAB*	yes	32% 24%	- -	Q09049 Q05780	[112]	1655703	
1h	HVO_0462 HVO_0461 (cont.)		yes	30% 27%	b0733 b0734	P0ABJ9 P0ABK2	[113]	6307994	
1h	NP_4296A NP_4294A	*coxA3* *coxB3*	yes	28% 33%	TTHA1135 TTHA1134	Q5SJ79 Q5SJ80	[114] [115]	2842747 7657607	
1i	HVO_2958 HVO_2959	*oadhAB1*			self	D4GY15 D4GY17	[116]	19910413	Ile indirectly assigned as substrate
1i	HVO_2958 HVO_2959 (cont.)				self		[117] [118] [119]	10832633 17571210 17906130	no substrate was identified; pyruvate and alphaKG excluded
1i	HVO_2595 HVO_2596	*oadhAB2*			self		[120] [119] [116]	12003954 17906130 19910413	no substrate was identified; pyruvate and alphaKG excluded
1i	HVO_0669 HVO_0668	*oadhAB3*			self		[119] [116]	17906130 19910413	no substrate was identified; pyruvate and alphaKG excluded
1i	HVO_2209	*oadhA4*			self				not yet analyzed experimentally
1i	HVO_2958 HVO_2959 (cont.)		yes/no	38% 52%	TA1438 TA1437	Q9HIA3 Q9HIA4	[121]	17894823	substrates are Ile, Leu, Val
1i	HVO_2595 HVO_2596 (cont.)		no	41% 41%	- -	Q57102 Q57041	[122]	1898934	substrate is acetoin
1i	HVO_2595 HVO_2596 (cont.)		unknown	40% 43%	BSU08060 BSU08070	O31404 O34591	[123]	10368162	substrate is acetoin
1i	HVO_0669 HVO_0668 (cont.)		unknown	54% 47%	BSU08060 BSU08070	O31404 O34591	[123]	10368162	substrate is acetoin
1i	HVO_0669 HVO_0668 (cont.)		unknown	49% 43%	- -	Q57102 Q57041	[122]	1898934	substrate is acetoin
1i	HVO_2209 (cont.)		unknown	38%	TA1438	Q9HIA3	[121]	17894823	substrates are Ile, Leu, Val

The column Section refers to the table listing the protein and to the section in the Results and in Appendix A. As an example, 2c covers topic (c) from the decimal-numbered Results Section 3.2. Amino Acid Biosynthesis. In Appendix A, this is covered under Appendix A. The corresponding proteins are listed in Table 2. For a few proteins, two sections are indicated (e.g., 1a/1b). The column Code refers to a haloarchaeal protein by its locus tag, which is mainly from *Haloferax volcanii* (HVO) but, also, from *Halobacterium salinarum* (OE), *Natronomonas pharaonis* (NP) and *Halohasta litchfieldiae* (halTADL). When the reconstruction of a complete pathway is presented, the unassigned genes are indicated as a “pathway gap”. In one case, we indicate the absence of a haloarchaeal ortholog by a dash. In the case of a complex, we either list more than one code or we list only one subunit together with the term (complex). All subunits of these complexes are listed groupwise in Appendix A. A protein may be shown in more than one row. From the 2nd row onwards, this is indicated by the term (cont.). The column Gene lists the assigned gene or a dash if no gene has been assigned. The assigned gene is only indicated in the first row of a protein. A set of four columns is used to relate a query protein to an experimentally characterized homolog, a GSP (Gold Standard Protein) (isofunc, %seq_id, Locus tag, UniProt). The column isofunc indicates if the query protein and its Gold Standard Protein homolog are isofunctional. The meanings of the terms used in this column in Table 1, Table 2, Table 3, Table 4, Table 5, Table 6, Table 7, Table 8 and Table 9 (yes, no, yes/no, probably, possibly, unclear, unknown, prediction, special and “-“) are described at the end of this legend. The column %seq_id indicates the protein sequence identity between the query protein and the homologous GSP. The column Locus tag contains the locus tag, if assigned. The column UniProt contains the UniProt accession of the GSP. GSPs are experimentally characterized as described in a publication. The column Reference links to the reference list of the manuscript. The column PMID lists the PubMed ID of the publication, if available. Otherwise, this is indicated as “not in PubMed”. Additionally, one PhD thesis is indicated (PhD_Mattar). The column Comment provides various types of additional information. The terms used in the column isofunc in Table 1, Table 2, Table 3, Table 4, Table 5, Table 6, Table 7, Table 8 and Table 9 have the following meanings: The term “yes” indicates that we consider the two proteins as isofunctional and annotate the query protein accordingly. The term “no” is used when we conclude that the proteins differ in function. Additional terms are used for more difficult cases. The term “yes/no” is used for GSPs that are multifunctional, and we assign only a subset of these functions to the query protein. The term “probably” is used when we consider it likely that the proteins are isofunctional and annotated the query protein accordingly (with the term probable added to the protein name). The term “possibly” is used when we see a good chance that the proteins are isofunctional but consider it too speculative to annotate the protein accordingly. The term “unclear” is used when we consider it likely that the same overall reaction is catalyzed but when reaction details, e.g., the energy-providing compound, are unresolved. The term “unknown” is used when it is not possible to predict the substrate of the query protein. The term “prediction” is used if a function assignment is based on bioinformatic analyses but not yet on an experimentally characterized homologous protein. The term “special” is used when multiple arguments have to be considered, with the full details provided in the corresponding section of Supplementary Text S1. Finally, a hyphen (“-“) is used when isofunctionality does not apply, e.g., when a homologous Gold Standard Protein could not be identified.

## Data Availability

Not applicable.

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
