# Peer review of "Open Issues for Protein Function Assignment in Haloferax volcanii and Other Halophilic Archaea"

_genes, 2021, doi:10.3390/genes12070963_

Round 1

Reviewer 1 Report

This manuscript summarizes previous works and lists open issues for protein function assignment in halophilic archaea, presenting hypotheses to explain the missing gaps in the metabolic pathway. The authors deeply scrutinized previous literatures, and their hypotheses are scientifically so interesting and reasonable to guide researchers to further studies on halophilic archaea.

Major points

Major point 1

Type of this manuscript should be "Review" rather than "Article" (original article) If the authors intend to publish this manuscript as "Article", it must be explicitly described in the manuscript what novel knowledge and evidences (neither speculations nor hypotheses) were obtained by this work for the first time.

Major point 2

The manuscript contains only one figure (and additional one in supplementary), and thus some arguments in text are difficult to understand without figures. For instance, Table 2 lists many genes with "Section 2a", readers cannot understand the presumed function of each gene product at all. Addition of a metabolic map like Figure 1 will help understanding of this section. The same is also true for Section 4a and others.

Major point 3

A part of contents of supplementary text is a duplicate of the main text. This makes the supplementary text redundant and can make readers confuse the significances of each text. For instance, L15-17 in Supp. Text ("In Haloferax, a conditionally lethal porAB mutant was unable to grow on glucose or pyruvate, but could grow on acetate. This demonstrates that alternative enzymes for conversion of pyruvate to acetyl-CoA do not exist in this species (Kuprat et al., 2021).") is a repetition of the main text (L233-236).

Minor points

L156

how ferredoxin Fdx is reoxidised

"reoxidised" is confusing because Fdx is oxidized (not reduced) during nitrate assimilation.

this might be achieved by the Nuo complex

The following paragraph describes that the haloarchaeal nuo complex lacks NADH-binding subunits. Thus, it should be explained why the authors can expect that the nuo complex can oxidize NADH without the necessary subunits.

L176-214

These paragraphs are redundant, and the issue is not clear.

Figure 1

The shape of the arrow of EC 1.3.1.76, which converts precorrin 2 to sirohydrochlorin, is confusing because it can be seen as two arrows. The arrow should be straight.

L350

Explanation or reference about "the cobalt-early pathway" should be added.

L375-390

Both (b) and (c) argue the same pathway gaps and seem to be redundant.

L391

Explanation of "late cobaltochelatase" should be added.

L456

No clear conclusion is drawn for HVO_0781. "The very strongly conserved gene neighbourhood with the NAD biosynthesis enzyme NadM may hint to an involvement in NAD biosynthesis" (Suppl. L980) seems to be a better conclusion.

L458

Section 6c does not appear in Table 6.

L544

Natrialbales,

Remove ",".

Figure S2

Figure S2 appears without Figure S1.

Author Response

Preliminary remark: after manuscript submission we became aware that we had left out a known open issue, the fact that two alternative enzymes exist for the last step in heme biosynthesis, while some haloarchaea lack a recognizable gene for the penultimate step. We have added this issue independent of any reviews. This also resulted in revision of Fig.1.

Major points

Major point 1

Type of this manuscript should be "Review" rather than "Article" (original article) If the authors intend to publish this manuscript as "Article", it must be explicitly described in the manuscript what novel knowledge and evidences (neither speculations nor hypotheses) were obtained by this work for the first time.

Response: We are aware that this manuscript is atypical but think that the “Review” category is inappropriate because a review summarizes current knowledge but is not intended to deal with “speculations and hypotheses” regarding gaps or likely errors in current knowledge. In identifying and exploring these gaps/errors, we performed multiple analyses (e.g. conservation of gene neighbourhood, ortholog distribution in haloarchaea, KEGG pathway analysis) to support our hypotheses/speculations. The results of these analyses provide “novel knowledge and evidence”, which is best categorized as research rather than a review. As we are not aware of a more suitable category that would best fit the nature of the manuscript, we hope that it will be accepted as an “Article”.

Major point 2

The manuscript contains only one figure (and additional one in supplementary), and thus some arguments in text are difficult to understand without figures. For instance, Table 2 lists many genes with "Section 2a", readers cannot understand the presumed function of each gene product at all. Addition of a metabolic map like Figure 1 will help understanding of this section. The same is also true for Section 4a and others.

Response: We agree and have created three additional figures in order to illustrate different aspects of the manuscript. One shows a comparison of tetrahydrofolate/methanopterin (section 5, Figure 2), one depicts lipid biosynthesis (section 7b, Figure 3), and one in the supplement , depicts a proposed purine degradation pathway (section 9g, Figure S2). The numbering of Suppl.Figures has been corrected in response to the last comment of this reviewer.

Major point 3

A part of contents of supplementary text is a duplicate of the main text. This makes the supplementary text redundant and can make readers confuse the significances of each text. For instance, L15-17 in Supp. Text ("In Haloferax, a conditionally lethal porAB mutant was unable to grow on glucose or pyruvate, but could grow on acetate. This demonstrates that alternative enzymes for conversion of pyruvate to acetyl-CoA do not exist in this species (Kuprat et al., 2021).") is a repetition of the main text (L233-236).

Response: we have attempted to provide an extensive description of topics in the supplement while presenting the same topics in a more condensed and concise way in the main text. It was our goal to avoid true duplications as much as possible. We are aware that the two versions about the conditional lethal porAB mutant are similar between supplement and main text but in the supplement we provide additional information about the positive control (growth on acetate). In the main text, we did not want to provide less information, because this experiment is the strongest argument that the information in KEGG is – in that exceptional case – simply incorrect. The same conclusion could have been drawn from the failed attempts to measure enzymatic activity, but a negative enzyme assay is a much less stringent proof for the absence of this enzymatic activity in the biological system. In general, we tried to “slim down” the main text as much as is possible without compromising clarity.

Minor points

L156: how ferredoxin Fdx is reoxidised

"reoxidised" is confusing because Fdx is oxidized (not reduced) during nitrate assimilation.

Response: We are very thankful to the reviewer for having caught this! The paragraph has been rephrased and now reads “It is yet unresolved how ferredoxin Fdx is reoxidised, but this might be achieved by the Nuo complex. This ferredoxin may well be involved in additional metabolic processes. In Hfx. volcanii, ferredoxin Fdx (HVO_2995) plays an essential role in nitrate assimilation (Zafrilla et al., 2011). However, in Hbt. salinarum this metabolic process for Fdx reoxidation does not exist.

L156 (cont.): this might be achieved by the Nuo complex

The following paragraph describes that the haloarchaeal nuo complex lacks NADH-binding subunits. Thus, it should be explained why the authors can expect that the nuo complex can oxidize NADH without the necessary subunits.

Response: We consider this a misunderstanding by the reviewer. In line 156, we deal with reoxidation of ferredoxin Fdx, not NADH. Reoxidation of Fdx is unlikely to depend on nuoEFG. In (d) we clearly state “Thus, the haloarchaeal nuo complex is unlikely to function as NADH dehydrogenase”. This is the opposite of stating “that the nuo complex can oxidize NADH without the necessary subunits”.

Additional note: Given that we consider the reviewer comment a misunderstanding, we tried to further improve the clarity of the text. We now more clearly state that the E.coli Nuo complex is a type I NADH dehydrogenase [in (c)], a term that is used in (d) and previously had not been well explained.

L176-214: These paragraphs are redundant, and the issue is not clear.

Response: We assume that the statement “paragraphs are redundant” refers to comparison of main text and of supplement. We did not detect any redundancy within each text. As stated above, we attempted to keep the main text “condensed and concise” but clearly deviate with respect to this issue for reasons detailed below. We are well aware that this is a complex issue. Thus, we reread the text with the reviewer’s statement “the issue is not clear” in mind. Unfortunately, we did not see how we could present this complicated issue even more clearly.

We believe that statements like “In our view, a paradigm shift is required. The obtained results call for a yet unanticipated novel mode of covalent heme attachment” are sufficiently speculative (and provocative) that they go well beyond what could ever be stated in a manuscript labelled Review. A bold claim like this also needs to be based on a clear and adequate coverage of the available information, and we have attempted to do this in the main text. The supplemental text goes further, and even provides experimental details about protein sequencing and gene cloning. This is done because the information is buried in a Ph.D. thesis which, to our knowledge, has never been made available in digital form and thus is difficult to access. Overall, we think that it is high time that experimentalists turn to this issue because “a yet unanticipated novel mode of covalent heme attachment” should not stay un-analyzed forever. We have to admit that this specific subject was one of the reasons why we decided to summarize our knowledge in form of an openIssues manuscript.

Figure 1: The shape of the arrow of EC 1.3.1.76, which converts precorrin 2 to sirohydrochlorin, is confusing because it can be seen as two arrows. The arrow should be straight.

Response: We agree. The figure has now been revised according to the reviewer’s suggestion. For additional modifications of the same figure, see the preliminary remark above.

L350: Explanation or reference about "the cobalt-early pathway" should be added.

Response: We have added two sentences to explain this pathway.

L375-390: Both (b) and (c) argue the same pathway gaps and seem to be redundant.

Response: we disagree with this statement. (b) deals with the pathway upstream of cob(II)yrinate a,c diamide, while (c) deals with the pathway downstream of this intermediate. It should be noted that the cobalt-early and cobalt-late pathways merge at cob(II)yrinate a,c diamide, so that pathway separation at this intermediate is considered biologically meaningful.

L391: Explanation of "late cobaltochelatase" should be added.

Response: We have reworded the section on the "late cobaltochelatase".

L456: No clear conclusion is drawn for HVO_0781. "The very strongly conserved gene neighbourhood with the NAD biosynthesis enzyme NadM may hint to an involvement in NAD biosynthesis" (Suppl. L980) seems to be a better conclusion.

Response: In our attempt to make the main text more “condensed and concise”, we shortened that conclusion as “Thus, HVO_0781 is a candidate to also be involved in NAD biosynthesis”. However, due to the subsequent sentence, it was not evident that this is the main conclusion of this paragraph. We have revised the paragraph so that it terminates with the original conclusion.

L458: Section 6c does not appear in Table 6.

Response: The proteins from section 6c had invalidly been labeled 6b. This has been corrected.

L544: “Natrialbales,” Remove ",".

Response: this has been corrected

Figure S2: Figure S2 appears without Figure S1.

Response: We have renumbered supplementary figures and tables starting with S1.

Reviewer 2 Report

I have read the manuscript with interest, and I think it is a very important and detailed study to point out problems associated with annotation in genomics. In particular, the analyzed system Haloferax volcanii is an important haloarcheon model and so the results should pay the way to improve the annotation approaches for halophilic archaea. 

Author Response

Response: thanks for this positive comment

Reviewer 3 Report

In this manuscript, the authors argue that the assignment of novel protein function can be unreliable if only based on sequence homology to Gold Standard Protein (GSP). They provide many examples that the function assignment of Haloferax volcanii proteins with GSP could be misleading without considering the biological context. Many of the arguments are not evidence-based and somewhat speculative. However, they have collected many examples and logical reasoning from literature to support their speculation. This manuscript could serve as a good reference for the Haloferax research community to look up the caveat of studied protein between the bioinformatic functional assignment and the actual function. The references in this manuscript are up-to-date and appropriate. Overall the reviewer thinks this article provides valuable inputs in the Haloferax community. 

Major comments:

  1. The authors provide hypotheses for the function of different Hfx. volcanii proteins. If those hypotheses are true, how would the author expect to add those results to the existing databases? 
  2. What are the components that are still lacking in the current databases but could be added to consider the organism's biological context?

Minor comments:

  1. “Nuo” complex should be “nuo” complex (Line 157)
  2. “SdhD” should be “sdhD” (Lines 195, 203, and 211)
  3. Hfx, volcanii” should be “Hfx. volcanii” (Lines 290, 584)

Author Response

Major comments:

1. The authors provide hypotheses for the function of different Hfx. volcanii proteins. If those hypotheses are true, how would the author expect to add those results to the existing databases? 

Response: if experimentally confirmed, the results will be published in regular papers and these will be incorporated into public databases. We are actively supporting databases in this endeavor by providing feedback via the available systems. If experimental analysis shows that an alternative hypothesis applies, disproving the hypothesis put forward by us, then this will also be published in regular papers and will make its way into the databases. If our hypothesis is simply disproven (e.g. the authors cannot measure the activity which we have predicted), the experiments may go unpublished. Such cases will, however, not have an impact on annotations, because our genome annotations are cautious and are not shaped according to our hypotheses.

2. What are the components that are still lacking in the current databases but could be added to consider the organism's biological context?

Response: we are not aware of any components still lacking for which a solution might be available, otherwise we would have mentioned them in the current manuscript.

Minor comments:

1. “Nuo” complex should be “nuo” complex (Line 157)

Response: a complex is built from proteins, and it is standard to capitalize gene names when not referring to the gene but to the encoded protein. Writing “Nuo complex” is an adaptation to that rule.

2. “SdhD” should be “sdhD” (Lines 195, 203, and 211)

Response: in all three cases, we refer to the encoded protein, not to the gene. Thus, it seems correct to use the capitalized form.

3. “Hfx, volcanii” should be “Hfx. volcanii” (Lines 290, 584)

Response: this has been corrected

Reviewer 4 Report

This manuscript analyzes the confounding problem of protein function assignment from genomic data and the usefulness/limitations of databases. Using a haloarchaeal model, Hfx. volcanii, the authors chose protein function assignments that were likely incorrect to explore and re-analyze. The results highlight annotation issues and a dissects common methods in protein functional predictions, ultimately serving a higher purpose of creating resources/methods for the larger scientific community. The Gold Standard Protein (GSP) based annotation strategy is an excellent approach and helps point out the problems that may be typical in genomic assignments of protein function. The Supplementary Files are well organized and help guide the reader through the specifics of the analyses. I support this work and it’s presentation here, and I have only a couple of suggestions that might it more accessible to the reader. 

Line 67: Consider revising, the word “insecure” seems inappropriate.

Line 129: Paracoccus species? spp?

Section 2: Materials and Methods. I do not feel your methods (e.g. GSP-based annotation strategy) of analysis are described here. I suggest a 2.3 section on methods that describe how you utilized the resources (described in section 2.1 and 2.2) for your work. Alternatively, you could add some sentences to describe this in sections 2.1 and 2.2, but I think it would be nice to pull it out into its own section in the methods. I know you describe this in detail in results, but maybe some of that could be moved here to methods.

Section 3: Results. I think the paragraph at the beginning “Open issues are organised below…” is not necessary since your organization below is so clearly demarcated by headers. Instead, I recommend a couple sentences regarding why you chose these pathways/protein families for your analysis or what led you to these particular categories.

Author Response

Line 67: Consider revising, the word “insecure” seems inappropriate.

Response: we replaced “insecure” by “uncertain”

Line 129: Paracoccus species? spp?

Response: We added the species name (denitrificans).

Section 2: Materials and Methods. I do not feel your methods (e.g. GSP-based annotation strategy) of analysis are described here. I suggest a 2.3 section on methods that describe how you utilized the resources (described in section 2.1 and 2.2) for your work. Alternatively, you could add some sentences to describe this in sections 2.1 and 2.2, but I think it would be nice to pull it out into its own section in the methods. I know you describe this in detail in results, but maybe some of that could be moved here to methods.

Response: We have now added a small paragraph in 2.1 where we briefly describe what we mean by Gold Standard Protein based genome annotation.

Section 3: Results. I think the paragraph at the beginning “Open issues are organised below…” is not necessary since your organization below is so clearly demarcated by headers. Instead, I recommend a couple sentences regarding why you chose these pathways/protein families for your analysis or what led you to these particular categories.

Response: we feel that it is helpful to briefly introduce the organization, because the demarcation may not yet be obvious at the very beginning of the results. Also, the list of topics is not available “at a glance”. Finally, we cannot be certain that the journal’s production team does not overturn our demarcations.

Response: why we chose these? These are all that we have come across in the more than 10 years of looking into this genome (at least all those that we identified over that time). We have added one sentence to state this, and we have added another sentence which explains how we restricted the scope of this manuscript.

Round 2

Reviewer 1 Report

All of the issues I previously pointed have been solved. I hope this paper will serve as a new foundation for studies on haloarchaea.